# LocationReasoner: Evaluating LLMs on Real-World Site Selection Reasoning

## Abstract

Recent advances in large language models (LLMs), particularly those enhanced through reinforced post-training, have demonstrated impressive reasoning capabilities, as exemplified by models such as OpenAI o1 and DeepSeek-R1. However, these capabilities are predominantly benchmarked on domains like mathematical problem solving and code generation, leaving open the question of whether such reasoning skills generalize to complex real-world scenarios. In this paper, we introduce *LocationReasoner*, a benchmark designed to evaluate LLMs' reasoning abilities in the context of real-world *site selection*, where models must identify feasible locations by reasoning over diverse and complicated spatial, environmental, and logistic constraints. The benchmark covers carefully crafted queries of varying difficulty levels and is supported by a sandbox environment with in-house tools for constraint-based location search. Automated verification further guarantees the scalability of the benchmark, enabling the addition of arbitrary number of queries. Extensive evaluations on real-world site selection data from Boston, New York, and Tampa reveal that state-of-the-art reasoning models offer limited improvement over their non-reasoning predecessors in real-world contexts, with even the latest OpenAI o4 model failing on 30% of site selection tasks. Moreover, agentic strategies such as ReAct and Reflexion often suffer from over-reasoning, leading to worse outcomes than direct prompting. With key limitations of LLMs in holistic and non-linear reasoning highlighted, we release LocationReasoner to foster the development of LLMs and agents capable of robust, grounded reasoning in real-world decision-making tasks. Codes and data for our benchmark are available at `https://anonymous.4open.science/r/LocationReasoner-DC5D`.

## 1 Introduction

Large language models (LLMs) have achieved tremendous success across a broad spectrum of tasks, ranging from human-like dialogue (Achiam et al., 2023; Team et al., 2023; Liu et al., 2024; Bai et al., 2023; GLM et al., 2024; Touvron et al., 2023) to agent-based simulations (Park et al., 2023; Shanahan et al., 2023; Wang et al., 2024a)—progress largely driven by data scaling and computation (Sutton, 2019; Wei et al., 2022a). Recent advances extend the notion of scaling beyond training, introducing test-time scaling techniques such as chain-of-thought (CoT) prompting (Wei et al., 2022b) and agentic strategies such as ReAct (Yao et al., 2023b), which enable deeper reasoning through intermediate steps and tool use. In particular, large-scale reinforcement learning has been applied during post-training to generate high-quality reasoning traces (e.g., CoT), giving rise to a new class of reasoning models such as OpenAI o1 (Jaech et al., 2024) and DeepSeek-R1 (Guo et al., 2025). These reasoning capabilities are critical for enabling LLMs to tackle complex real-world tasks that often require decomposing problems, exploring alternative strategies, and interacting with external tools.

Benchmarks play a crucial role in driving LLM progress (Patterson, 2012), and the rapid advancement of these models has created a growing demand for more challenging and informative evaluation tasks. While general language understanding benchmarks are widespread (Chiang et al., 2024; Wang et al., 2024b; Hendrycks et al., 2021a; Rein et al., 2024; Huang et al., 2023), assessing reasoning capabilities remains considerably more difficult, primarily due to the challenge of verifying the correctness of LLMs' reasoning outputs. As a result, current reasoning models are predominantly evaluated on domains with easily verifiable solutions, where verifying a solution is much simpler

than generating one. This includes domains such as mathematical problem solving and programming, exemplified by benchmarks such as AIME (MAA, 2024), MATH500 (Hendrycks et al., 2021b), and CodeForces (Quan et al., 2025). In contrast, few benchmarks have tackled real-world reasoning scenarios, which often involve more complicated constraints. For instance, the TravelPlanner benchmark (Xie et al., 2024) evaluates LLM agents on practical planning tasks but does not include recent reasoning models and lacks automated verification mechanisms, relying heavily on manual annotation. Consequently, it remains unclear whether current reasoning LLMs and agents can effectively generalize to real-world reasoning tasks. This gap highlights the urgent need for a new benchmark that targets real-world reasoning while remaining scalable and annotation-free.

In this work, we introduce *LocationReasoner*, a benchmark designed to evaluate the real-world reasoning capabilities of LLMs through the task of *site selection*. Site selection is a common decision-making process in practical business settings, where the objective is to identify feasible locations based on a diverse set of criteria. This task presents a significant reasoning challenge, as it requires LLMs to perform structured location search and filtering over queries that involve multiple constraints with complex logical dependencies. For instance, selecting a site for a new restaurant may demand reasoning over temporal consumption constraints and spatial transportation constraints, with various conditions often combined through Boolean logic. To comprehensively assess LLMs' ability to reason under such practical and multifaceted conditions, we construct the LocationReasoner benchmark, which includes site selection queries spanning a wide range of difficulty levels. The benchmark is supported by a sandbox environment equipped with offline datasets and in-house tools for constraint-based location search.

LocationReasoner differs fundamentally from existing benchmarks in two key aspects. First, it focuses on practical reasoning, where LLMs must decompose complex constraints into smaller executable steps and utilize available tools to solve each query. Second, the proposed benchmark features deterministic and automatically verifiable queries, enabling fully scalable evaluation without human annotation, and allowing seamless extension to much larger benchmark suites.

The key contributions of this work are: (1) We introduce LocationReasoner, a real-world reasoning benchmark that grounds LLMs in practical site selection tasks. The benchmark includes a suite of curated datasets, toolsets, and a sandbox environment to enable out-of-the-box evaluation of LLMs in complex constraint-based reasoning. (2) We conduct a comprehensive evaluation of four major LLM families,—OpenAI (Jaech et al., 2024), Gemini (Team et al., 2023), Claude (Anthropic, 2024), and DeepSeek (Guo et al., 2025) —covering both general-purpose and reasoning-augmented models, and two representative agentic workflows, ReAct (Yao et al., 2023b) and Reflexion (Shinn et al., 2023). (3) Our benchmarking results reveal that current LLMs struggle to effectively handle real-world reasoning challenges. For instance, the latest OpenAI o4 model achieves only 69.99% success on site selection queries. Furthermore, reasoning-augmented models and agentic strategies provide limited gains compared to direct prompting of general models. In-depth analysis uncovers key limitations in holistic and nonlinear reasoning, pointing to critical bottlenecks that need to be addressed to improve LLM performance on practical real-world tasks.

## 2 LOCATIONREASONER

We center the LocationReasoner benchmark around practical site selection, a problem that is easy to understand and intuitive, but also requires multi-step reasoning across all constraints. As illustrated in Figure 1, we construct the entire query generation and execution pipeline so that it can be fully automated to support scalable evaluation. Queries are generated through rule-based and LLM-based approaches to cover diverse test scenarios, using real data from Boston, New York, and Tampa. A fixed set of in-house tools are provided to achieve a controlled and interpretable environment for testing the proposed LLMs' ability to reason and plan under real-world constraints. The queries are routed through two execution pathways: a deterministic code-based system that applies filters using in-house tools, and an LLM agent system that interprets and solves the same

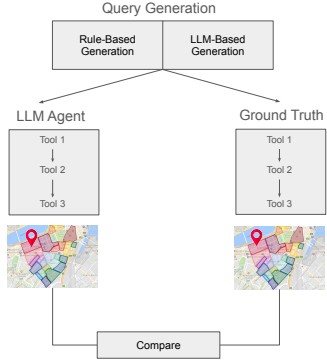

Figure 1: Execution pipeline.

Table 1: In-house tools description.

| Category | Tool Name | Description |
|----------|-----------|-------------|
| Loader Tools | get_poi_spend_dataset | Loads POI metadata and consumer spending patterns. |
| | get_parking_dataset | Loads parking facility data including location and area. |
| Zone Tools | create_zone | Creates a zone DataFrame of POIs grouped by zone. |
| | assign_parking_zones | Assigns each parking lot to a zone. |
| | get_zone_center | Returns the geographic center of a zone. |
| | get_neighbor_zones | Finds the closest zones to a given zone by distance. |
| Analysis Tools | get_spendparam_years | Computes aggregated spend over selected years. |
| | get_num_parking | Counts parking lots per zone. |
| | get_largest_parking_lot_area | Returns the area of the largest parking lot per zone. |
| | get_largest_parking_capacity | Returns the maximum parking capacity per zone. |
| | get_distance_km | Calculates haversine distance between two points. |
| Filter Tools | filter_df_based_on_zone | Filters a DataFrame to only include specified zones. |
| | filter_pois_by_top_category | Filters POIs by top-level category. |
| | filter_pois_by_sub_category | Filters POIs by sub-category. |
| | get_transport_pois_in_zone | Finds transport POIs by type and groups them by zone. |
| Population Tool | get_population | Retrieves population per zone using Google Places API. |

queries. The system logs and compares the final site selections from both paths, enabling large-scale benchmarking without manual annotation or intervention. It is worth noting that LocationReasoner supports fully automated query generation and verification, thus the benchmark can be easily scaled up to incorporate an arbitrary number of queries.

## 2.1 ENVIRONMENT SETTING

We construct a sandbox environment to ensure a stable and consistent evaluation of all LLM agents, where datasets are fixed, no external API calls are made, and all tool functionalities remain constant.

**Datasets.** The benchmark database is constructed by integrating multiple real-world datasets to provide a rich and realistic foundation for constraint-based planning. Specifically, we construct the database by merging data from the SafeGraph dataset[1], which includes detailed information on Points of Interest (POIs), parking facilities, and consumer spending patterns from 2019 to 2025. To enrich the spatial and demographic context, we incorporate population data from Google Places API[2] and transportation network data from OpenStreetMap[3]. This integrated dataset enables agents to reason about accessibility, mobility, and demand when evaluating site suitability.

**Tools.** To support structured reasoning, the sandbox contains a set of in-house tools that LLM agents can call during problem solving. These tools expose curated functionalities such as filtering zones, analyzing parking availability, retrieving spending patterns over time, and evaluating transportation accessibility. We provide a detailed description of each tool in Table 1.

## 2.2 QUERY DESIGN

We design queries to systematically evaluate an LLM agent's ability to reason over various constraints using in-house tools. By defining constraints, threshold ranges, and logical compositions, queries can be generated in bulk through either rule-based generation or LLM-based generation. The rule-based approach is rigid and syntactically precise, as predefined constraints are directly encoded into executable templates. It ensures that all conditions are logically well-formed and interpretable by deterministic code. LLM-based generation introduces linguistic diversity. The system feeds the same set of structured constraints to a language model and asks it to produce free-form queries in natural language. Queries are categorized into three difficulty levels, reflecting an increasing level of complexity. Specifically, difficulty is determined by the number of constraints, diversity of tools involved, the depth of reasoning required, the linguistic complexity of the query, and the necessity to synthesize and compare heterogeneous data sources across spatial and temporal dimensions. The difficulty level enables controlled benchmarking across different kinds of reasoning scenarios,

---

[1]https://www.safegraph.com/

[2]https://developers.google.com/maps/documentation/places/web-service/overview

[3]https://www.openstreetmap.org/

ranging from straightforward constraint filtering to complex, multi-faceted decision-making. The query variants mimic the types of natural, often nuanced questions that real users might ask when selecting a site, testing the LLM's ability to precisely extract and interpret user requests. We provide detailed descriptions and examples of different difficulty levels as follows:

- **Simple.** Simple queries involve direct calls to a single in-house tool with a numerical threshold or binary filter. These queries require minimal coordination between tools.
  *Example: "I want to build a retro roller rink with at least 3 parking lots. Where should I look?"*

- **Medium.** Queries at this level introduce logical dependencies between constraints and often require coordination across multiple tools. These queries combine 2 to 3 simple constraints using Boolean operators such as AND or OR, chaining together multiple easy-level conditions. Additionally, medium queries demand mathematical reasoning and often involve relationships across structural features (e.g., neighboring zones), socio-economic indicators (e.g., spending patterns), or temporal variation (e.g., comparing data from different years).
  *Example: "Find zones where at least 35% of total raw total spend in 2020 comes from the top category {Restaurants and Other Eating Places}."*

- **Hard.** Hard queries consist of typically 3-6 composite constraints derived from simple or medium-level logic, combined using AND, OR, and NOT operators. These queries demand nuanced, multi-tool reasoning, trade-off analysis, and often involve exclusionary conditions. They closely resemble real-world decision-making scenarios in site selection, where users express both strong preferences and explicit disqualifiers, requiring the agent to balance multiple objectives while avoiding infeasible options.
  *Example: "Launch a creative co-working café — needs at least 26 POIs total in that area and also strong local spending with 50%+ from sub-category of {Full-Service Restaurants} in 2022, but I'm not interested if the number of POIs in that same category dominates the area by 30%."*

## 2.3 REASONING APPROACHES

Our goal is to assess whether LLMs can reason over multi-step planning constraints, invoke appropriate in-house tools, and return correct site selection outputs. We explore three distinct strategies:

- **Direct Prompting.** We provide LLMs with relevant context of available in-house tools and structured data, prompting them to directly generate executable Python code that fulfills the query and outputs a list of candidate locations.

- **ReAct (Reasoning + Acting).** The ReAct framework (Yao et al., 2023b) is adopted to guide the agent through an iterative loop of Thought-Action-Observation (TAO). In the Thought step, the agent reflects on its current state and reasons about what to do next. In the Action step, it either calls an in-house tool or runs custom Python code. Executing custom Python code is particularly useful for performing transitional logic such as filtering results, combining intermediate outputs, or applying custom calculations. The resulting output is captured in the Observation step and stored as part of the agent's evolving internal state. These observations are continuously fed back to the model, enabling it to reason with an evolving memory of prior tool outputs and decisions. The model is also allowed to store and reference intermediate results.

- **ReAct + Reflexion.** Reflexion (Shinn et al., 2023) extends the ReAct framework by allowing the model to reflect on failed attempts such as code errors or reaching the maximum step limit without producing a result. In these cases, the model receives feedback from its prior reasoning trace and uses it to revise its approach. This added layer of self-correction helps the model identify mistakes, adjust its logic, and improve tool usage in subsequent retries, making it especially useful for handling complex, multi-constraint queries.

## 2.4 AUTOMATED VERIFICATION

To ensure that each test case has a clearly defined and feasible solution, we develop a set of deterministic, rule-based functions that serve as the ground truth logic for evaluating site selection. These logic scripts are constructed using the same in-house tools available to the LLM agent and are designed to satisfy all constraints specified in each query. For each query, the script returns a set of valid zones that fully meet the defined conditions. During evaluation, the output of the LLM agent is

compared directly against these ground truth results to assess correctness. This approach ensures consistent, reproducible evaluation across all queries, and is automated without any human annotation. We evaluate the agent's ability to generate complete outputs, satisfy user-defined constraints, and accurately identify valid zones with the following complementary metrics:

- **Delivery Rate** measures the proportion of queries for which the LLM agent successfully returns selected zones. A process is considered completed if the agent successfully produces an output without execution failure. For agents evaluated via direct prompting, the generated code must run without syntax or runtime errors. For agents evaluated under the ReAct or Reflexion framework, the agent must avoid generating incorrect or unexecutable code, prevent infinite reasoning loops, and successfully complete the task within a strict limit of 30 reasoning steps. This metric captures basic functional reliability, as failed plans cannot be evaluated for correctness.

- **Perfect Pass Rate** measures the percentage of outputs in which the set of zones returned by the agent exactly matches the ground truth derived from our objective logic. A plan is considered perfect only if it satisfies all constraints and produces no extraneous or missing zones. This strict metric ensures full compliance with the query's requirements and serves as a high-confidence indicator of planning accuracy.

- **Precision, Recall, and F1 Score** are standard classification metrics assessing zone-level selection accuracy in addition to binary success metrics.

## 3 BENCHMARKING RESULTS

For the direct prompting setup, we test a range of models spanning multiple providers, including DeepSeek (V3, R1), OpenAI (GPT-4o, o4-mini), Gemini (1.5, 2.5), and Claude (3, 3.5). Each model receives the same prompt structure, tool descriptions, and structured input data, and is tasked with generating executable Python code that satisfies the constraints specified in each query and outputs a list of candidate locations. Notably, we cover both general and reasoning models of each LLM family, to assess whether such advertised reasoning capabilities lead to improved performance in real-world structured, constraint-driven reasoning tasks. For the ReAct and Reflexion frameworks, we focus our evaluation on OpenAI's GPT-4o model. This setup allows us to examine the benefits of iterative reasoning, tool chaining, and self-correction over single-shot prompting approaches.

Table 2 summarize the results of 316 site selection queries (114 simple, 102 medium, 102 hard) using Boston data, from which we have the following observations.

**Limited overall performance.** The overall model performance on the LocationReasoner benchmark remains limited, as most models fail to generate complete and accurate outputs across varying difficulty levels. The average perfect pass rate for 8 different LLMs is only 49.61%. Even the best-performing model, *OpenAI o4-mini*, achieves only a 69.99% perfect pass rate and an overall F1 score of 0.57. This reflects a significant gap between current LLM capabilities and the demands of complex real world reasoning tasks.

**Difficulty sensitivity.** Across all models, performance consistently degrades as task difficulty increases. All key metrics decline from simple to medium to hard queries. Specifically, the average perfect pass rate on hard queries is only 33.82%, which is much lower than 61.49% of simple queries. This trend highlights that while some LLMs may be capable of handling basic filtering or single-constraint tasks, they struggle to generalize to multi-step reasoning and more compositional decision-making challenges.

**Reasoning LLMs offer significant improvements.** With the exception of the DeepSeek family, the other three families show substantial improvements in their reasoning models compared to their non-reasoning baseline counterparts. For example, *Gemini 2.5*, advertised as a reasoning-optimized model, achieves an overall F1 score of 0.48 and a perfect pass rate of 62%. Its predecessor, *Gemini 1.5*, reaches only a 27% perfect pass rate and an F1 score of 0.21. However, the gains remain far from transformative. *Gemini 2.5* still fails over 35% of simple queries and more than half of medium ones. This underscores that even when reasoning-optimized models do improve over their previous versions, their performance still falls short of the reliability required for real-world reasoning tasks.

**Agentic strategies do not guarantee better results.** Agentic strategies such as ReAct and Reflexion do not outperform direct prompting. Despite being designed to simulate step-by-step human reasoning,

Table 2: Performance metrics on LocationReasoner by model and difficulty level on Boston data.

| Model | Difficulty | Delivery Rate | Perfect Pass | Precision | Recall | F1 Score |
|---|---|---|---|---|---|---|
| **Direct Prompting** | | | | | | |
| **Deepseek V3** | Simple | 92.98% | 68.42% | 0.73 | 0.69 | 0.67 |
| | Medium | 79.17% | 60.42% | 0.55 | 0.52 | 0.51 |
| | Hard | 77.19% | 42.11% | 0.24 | 0.20 | 0.18 |
| | Overall | 83.33% | 57.00% | 0.51 | 0.47 | 0.46 |
| **Deepseek R1** | Simple | 92.59% | 75.00% | 0.73 | 0.71 | 0.69 |
| | Medium | 76.04% | 61.46% | 0.55 | 0.54 | 0.51 |
| | Hard | 71.57% | 36.27% | 0.23 | 0.25 | 0.22 |
| | Overall | 80.39% | 58.92% | 0.51 | 0.51 | 0.48 |
| **OpenAI 4o** | Simple | 89.81% | 71.30% | 0.66 | 0.61 | 0.60 |
| | Medium | 85.42% | 55.21% | 0.57 | 0.46 | 0.45 |
| | Hard | 88.24% | 37.25% | 0.26 | 0.17 | 0.15 |
| | Overall | 87.91% | 55.00% | 0.50 | 0.42 | 0.40 |
| **OpenAI o4-mini** | Simple | 91.82% | 81.82% | 0.74 | 0.74 | 0.72 |
| | Medium | 89.58% | 73.96% | 0.67 | 0.64 | 0.61 |
| | Hard | 88.24% | 53.92% | 0.42 | 0.40 | 0.38 |
| | Overall | 89.94% | 69.99% | 0.61 | 0.59 | 0.57 |
| **Gemini 1.5** | Simple | 46.30% | 29.63% | 0.32 | 0.30 | 0.30 |
| | Medium | 51.04% | 32.29% | 0.27 | 0.25 | 0.23 |
| | Hard | 46.08% | 19.61% | 0.13 | 0.08 | 0.09 |
| | Overall | 47.71% | 27.00% | 0.24 | 0.21 | 0.20 |
| **Gemini 2.5** | Simple | 92.59% | 77.78% | 0.65 | 0.65 | 0.64 |
| | Medium | 80.21% | 63.54% | 0.52 | 0.50 | 0.48 |
| | Hard | 83.33% | 45.10% | 0.33 | 0.35 | 0.32 |
| | Overall | 85.62% | 62.00% | 0.50 | 0.51 | 0.48 |
| **Claude 3 Haiku** | Simple | 66.67% | 35.19% | 0.39 | 0.34 | 0.31 |
| | Medium | 47.92% | 21.88% | 0.14 | 0.08 | 0.07 |
| | Hard | 30.39% | 9.80% | 0.03 | 0.01 | 0.01 |
| | Overall | 48.69% | 23.00% | 0.19 | 0.15 | 0.13 |
| **Claude 3.5 Haiku** | Simple | 67.59% | 52.78% | 0.51 | 0.50 | 0.48 |
| | Medium | 65.62% | 54.17% | 0.43 | 0.40 | 0.39 |
| | Hard | 46.08% | 26.47% | 0.22 | 0.21 | 0.18 |
| | Overall | 59.80% | 44.00% | 0.39 | 0.37 | 0.35 |
| **Agentic Workflow** | | | | | | |
| **ReAct** | Simple | 72.22% | 39.81% | 0.44 | 0.35 | 0.36 |
| | Medium | 92.71% | 26.04% | 0.37 | 0.17 | 0.18 |
| | Hard | 85.29% | 16.67% | 0.25 | 0.07 | 0.09 |
| | Overall | 83.01% | 28.00% | 0.35 | 0.20 | 0.21 |
| **ReAct + Reflexion** | Simple | 99.07% | 32.41% | 0.54 | 0.27 | 0.29 |
| | Medium | 94.79% | 37.50% | 0.52 | 0.29 | 0.31 |
| | Hard | 97.06% | 29.41% | 0.31 | 0.13 | 0.14 |
| | Overall | 97.06% | 33.00% | 0.46 | 0.23 | 0.24 |

*ReAct + Reflexion* achieves only a 33% perfect pass rate and an overall F1 score of 0.24, which is substantially lower than the best-performing LLMs using direct prompting. This suggests that agentic strategy alone is insufficient without more reliable reasoning abilities of the base model. We provide detailed explanations on the inferior performance of agentic strategies in Appendix A.2.

## 4 SCALABILITY AND STATISTICAL ROBUSTNESS

The automated query generation and verification property of LocationReasoner ensures its scalability to incorporate more queries and more datasets. In this section, we re-ran the experiments in two additional cities: New York and Tampa. These cities feature distinctive urban layouts and provide a more comprehensive testbed for spatial reasoning. From Table 4 in Appendix we observe that the results on these new datasets are consistent with our initial findings, showing that even top-performing models struggle with complex spatial reasoning, and highlighting the persistent challenges in this

domain. Particularly, the best-performing model, Gemini 2.5, can only achieve a Perfect Pass Rate of 58.40% on hard queries in New York, and 61.74% in Tampa. This expansion demonstrates that our framework is robust and applicable to different site selection environments.

To evaluate the robustness of our benchmark with respect to dataset size, we further conducted an analysis tracking the Perfect Pass Rate of models as the number of queries increases using the Boston dataset. As shown in Figure 2, the performance for all models begins to converge around 150 queries, with subsequent fluctuations remaining minimal. This indicates that our choice of ~300 queries per city is more than sufficient to provide a stable and statistically significant signal of model capability. Furthermore, since our queries and the sandbox environment support fully automated evaluation, the size of the benchmark can be easily and efficiently increased in future work to test even more fine-grained aspects of LLM reasoning. This focus on automated, low-cost scalability makes LocationReasoner a uniquely practical and sustainable tool for rigorously testing complex, constraint-based spatial and logical reasoning that is complementary to the tested abilities on other benchmarks.

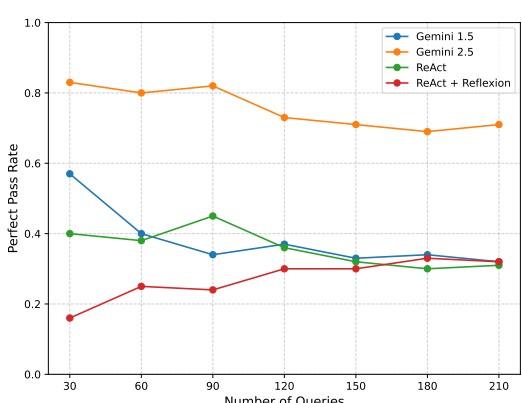

Figure 2: Model performance convergence with increasing query count.

## 5 ERROR ANALYSIS

LLMs consistently struggle with five primary failure types, and we show the breakdown of these failures in Figure 3. These errors often overlap, but their root causes reflect distinct reasoning weaknesses that hinder constraint satisfaction and plan validity.

**Logic Error** encompasses broken reasoning chains, incorrect constraint handling, and improperly ordered filtering logic. This includes sequential filtering mistakes where each constraint is applied too early, altering the input for future conditions. In ReAct, logic errors often emerge as inconsistent multi-step chains, where prior outputs are forgotten or overwritten. In direct prompting, they tend to surface as incorrect condition checks or missing final evaluations.

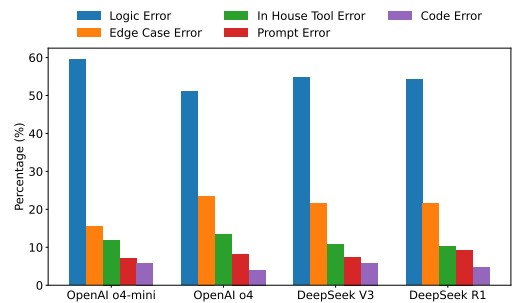

Figure 3: Breakdown of failure types for OpenAI and DeepSeek models

**Edge Case Error** indicates that LLMs frequently misinterpret constraints that require careful logic around null values or boundary conditions. A common example involves requests like "fewer than 3 competitors," where agents exclude zones with zero competitors–despite these being valid. These failures stem from inflexible logic structures and an inability to reason inclusively.

**Tool Error** captures failures where the model misuses, miscalls, or entirely ignores in-house tools. Examples include passing incorrect argument formats, selecting a tool misaligned with the constraint type, or chaining incompatible tool outputs. LLMs sometimes bypass the toolset and attempt to write their own versions of utility functions, which introduce further bugs and inconsistencies. These issues reveal brittle tool affordance understanding and poor mapping between language and tool usage.

**Prompt Error.** Prompt-related failures involve misinterpreting user intent. In many cases, they misclassify the type of reasoning needed (e.g., assuming a ranking is needed when only filtering is requested), leading to logic or tool selection failures. These mistakes reflect weak grounding in language understanding and sensitivity to phrasing variations.

Table 3: Correction rates (%) for reasoning vs. non-reasoning model pairs across difficulty levels.

| Difficulty | OpenAI o4-mini vs 4o | Claude 3.5 vs 3 | Deepseek R1 vs V3 | Gemini 2.5 vs 1.5 | ReAct Reflexion vs None |
|---|---|---|---|---|---|
| **Simple** | 18.89 | 28.26 | 10.10 | 21.74 | 6.67 |
| **Medium** | 23.68 | 30.30 | 2.82 | 20.00 | 18.82 |
| **Hard** | 17.95 | 16.67 | 4.08 | 21.62 | 16.87 |

**Code Error.** Code-specific issues include syntax errors, missing return statements, incorrect indentation, and unsafe operations such as division by zero. These errors prevent valid plans from executing altogether and reduces the agent's delivery rate.

We evaluate whether reasoning models can correct the errors made by their non-reasoning counterparts. Table 3 shows the results where *Correction Rate* is defined as the proportion of test cases in which the reasoning model produces a correct output while the non-reasoning model fails. Our results indicate that reasoning models are particularly effective in correcting failures in medium queries. This is likely because medium queries contain a high concentration of logic-related errors, which are well suited to correction through enhanced reasoning. In contrast, easy queries tend to involve edge cases, while hard queries are often dominated by syntax issues and execution errors, where reasoning alone is insufficient to guarantee success. For example, OpenAI's o4-mini corrects 25.36% more errors over GPT-4o in medium queries than hard or simple ones. However, the extent of improvement varies considerably across model families, highlighting that the effectiveness of reasoning depends not only on the presence of reasoning mechanisms but also on how well they are integrated and executed. We provide concrete failure examples including the generated codes as well as prompt ablation experiments in Appendix A.3-A.4.

## 6 CASE STUDY

We observe a high frequency of logic errors across all models and test cases. Therefore, we focus our analysis on this category. Below are representative test cases that specifically highlight different forms of logical failure.

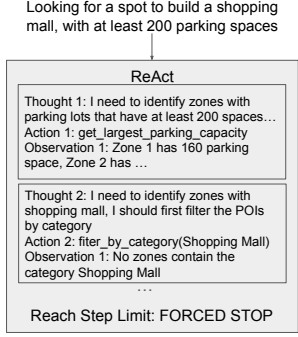

**Over-reasoning (Direct Prompting and ReAct).** Agents frequently demonstrate over-reasoning by performing unnecessary operations that go beyond the requirements of the prompt. This often stems from misinterpreting implicit cues. In direct prompting, agents tend to apply extra filters that overly constrain the result set, introducing conditions not specified in the query. In the ReAct setting, the model continues to invoke tools and apply filters even after all constraints have been met, lacking a clear signal for termination. As illustrated in Figure 4, the

Figure 4: Over-reasoning

ReAct agent first retrieves parking capacity correctly, then unnecessarily proceeds to filter for the presence of shopping malls and many other unrelated tools. The agent accumulates irrelevant actions until it reaches the step limit and fails to return an answer, despite having the correct information early on.

**Sequential Filtering (Direct Prompting).** In complex queries involving multiple chains of constraints, the model often applies filters sequentially rather than evaluating the logic holistically. As shown in Figure 5, the query contains an OR condition between competitor count and population. The correct approach (right) evaluates both branches independently before merging the results. However, the LLM agent (left) applies filters sequentially, first by competitor count and then by population, prematurely discarding zones that could have satisfied the second

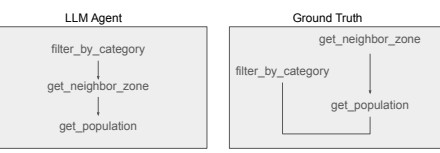

Figure 5: Sequential Filtering

count and then by population, prematurely discarding zones that could have satisfied the second

condition. This flawed execution leads to valid zones being excluded, revealing a logical failure where sequential filtering overrides the intended logical structure.

## 7 RELATED WORK

**LLM Reasoning.** Reasoning represents a key capability of advanced LLMs, enabling them to decompose tasks into smaller subproblems, perform structured search, and make decisions for complex problem solving. Various techniques have been proposed to enhance the reasoning abilities of LLMs. Prompting methods (Wei et al., 2022b; Yao et al., 2023a; Besta et al., 2024), such as chain-of-thought (CoT) prompting (Wei et al., 2022b), encourage models to generate intermediate and structured reasoning steps before producing final answers. These sequential reasoning patterns can be further extended into more expressive structures, such as trees (Yao et al., 2023a), for handling more complex reasoning workflows. Beyond prompting, recent research has shown that reasoning capabilities can also be distilled into LLMs' generation itself through post-training strategies (Guo et al., 2025; Snell et al., 2024; Brown et al., 2024; Zelikman et al., 2024). In particular, reinforcement learning is often used to fine-tune models to produce high-quality reasoning trajectories, such as CoT-style paths (Guo et al., 2025; Havrilla et al., 2024; Kumar et al., 2024; Carta et al., 2023). In parallel, agentic approaches offer an alternative paradigm, equipping LLMs with tools, external memory, and planning mechanisms to solve complex tasks (Wiesinger et al., 2024; Yao et al., 2023b; Shinn et al., 2023; Zhao et al., 2024; Shen et al., 2023; Schick et al., 2023). For instance, the ReAct framework (Yao et al., 2023b) enables in-context learning by prompting models to engage in iterative thought-action-observation cycles. In this work, we systematically evaluate both LLMs and reasoning-enhanced strategies on practical real-world reasoning tasks through our proposed benchmark.

**Benchmarks for LLMs and Agents.** Benchmarks play an essential role in the development and evaluation of LLMs. Fundamental language capabilities are typically assessed through general-purpose benchmarks targeting language understanding and factual question answering (Srivastava et al., 2022; Wang et al., 2024b;b; Rein et al., 2024). In addition, domain-specific benchmarks have been introduced to evaluate LLMs' proficiency in specialized areas such as biology (Chen & Deng, 2023) and law (Fei et al., 2023). As reasoning has emerged as a central focus in recent LLM research, benchmarks tailored specifically to reasoning have become increasingly important. Given the complexity of evaluating LLM reasoning, existing benchmarks often focus on domains with well-defined verifiers, where checking the correctness of a solution is significantly easier than generating it. Two such domains–mathematics and programming–have become prominent battlegrounds for reasoning-oriented LLMs (Hendrycks et al., 2021b; Quan et al., 2025; MAA, 2024), exemplified by benchmarks such as MATH500 (Hendrycks et al., 2021b) and CodeForces (Quan et al., 2025). However, constructing benchmarks for practical reasoning tasks remains challenging, largely due to the difficulty of defining objective metrics and the high cost of manual annotation (Xie et al., 2024). In contrast, our benchmark evaluates LLMs and agents on real-world site selection reasoning while maintaining full scalability through automatic verification, requiring no human annotation. This enables robust, reproducible, and extensible evaluation of reasoning capabilities in realistic decision-making contexts.

## 8 CONCLUSION

In this paper, we present LocationReasoner, a benchmark designed to evaluate LLMs and agents on real-world site selection reasoning tasks, supported by a sandbox environment enriched with built-in tools and curated datasets. Through systematic evaluation of both general-purpose and reasoning-oriented LLMs from multiple providers, along with various agentic strategies, we find that current models perform poorly, with even the most advanced reasoning LLM failing to solve over 30% of the queries. Our analysis identifies key bottlenecks, particularly in over-reasoning and the challenges of sequential problem-solving, which limit the effectiveness of LLMs in complex real-world reasoning scenarios. LocationReasoner serves as a scalable and extensible testbed: new constraints can be added and automatically validated without the need for human annotation. We hope LocationReasoner will drive progress in enhancing the reasoning capabilities of LLMs and contribute to the advancement of general intelligence for solving practical, high-stakes problems.

ETHICS STATEMENT

The development of the LocationReasoner benchmark utilizes publicly available and commercially licensed datasets, including SafeGraph, OpenStreetMap, and Google Places API. The data is aggregated and anonymized, focusing on points of interest, consumer spending patterns, and geographical zones, thereby minimizing risks to individual privacy. We acknowledge that the task of site selection, while presented here as a testbed for LLM reasoning, has real-world societal and economic implications. Tools that optimize location-based decisions can influence local economies and potentially exacerbate existing socio-economic disparities. The underlying datasets may reflect historical biases in urban development and commercial activity. Our benchmark does not attempt to mitigate these inherent data biases but rather provides a framework to evaluate how models reason with complex, real-world data as it exists. By releasing this benchmark, we aim to encourage research into developing more robust and grounded AI reasoning systems, and we hope it will enable the community to further study the fairness and societal impact of such systems in decision-making contexts.

REPRODUCIBILITY STATEMENT

We are committed to ensuring the reproducibility of our research. All code, datasets, query generation scripts, and the evaluation suite for the LocationReasoner benchmark are made available in an anonymized repository at https://anonymous.4open.science/r/LocationReasoner-DC5D. The repository includes the complete sandbox environment, the implementation of all in-house tools, and the deterministic, rule-based functions used to establish ground truth for automated verification. Our experiments were conducted using commercially available LLMs accessed via APIs, including models from the OpenAI, Gemini, Claude, and DeepSeek families. The paper provides a detailed description of our experimental setup, including reasoning approaches (Direct Prompting, ReAct, and Reflexion), prompt structures, and the specific metrics used for evaluation, enabling the community to fully replicate our findings and extend this work.

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

## A  APPENDIX

### A.1  USE OF LLMS

LLMs are used for polishing and proofreading the text. All core ideas were independently conceived by the authors.

### A.2  REASONING ARCHITECTURE COMPARISON

Direct code generation consistently outperforms ReAct/Reflexion because of the fundamental differences in how each paradigm approaches logical reasoning. Below, we highlight two critical architectural dimensions: holistic vs. chunked planning and linear vs. nonlinear reasoning.

**Holistic vs. Chunked Planning**  Code generation promotes holistic planning. The model sees the entire query and designs a solution that integrates all constraints into one coherent logic block. It defers execution until the entire logical structure is assembled and performs filtering or aggregation after all conditions are accounted for. On the contrary, ReAct operates in chunks where each TAO cycle is focused on a single subproblem. The agent reasons incrementally, using only its most recent observations to decide what to do next. While this can help with short-term reactivity, the model loses sight of the broader goal. Instead of building a global solution from the outset, ReAct focuses on local decisions by evaluating one tool at a time without fully understanding how each TAO interacts, which results in fragmented logic, redundant actions, and plans that satisfy isolated constraints but fail as a cohesive whole.

**Nonlinear vs. Linear Reasoning**  Direct code generation follows a nonlinear reasoning approach. The model has the flexibility to write logic in any order, reuse variables fluidly, and jump between intermediate steps as needed. This flexibility allows the agent to integrate information from different sources without being tied to a rigid step-by-step flow. In contrast, ReAct enforces a linear reasoning pattern. The agent processes the query as a series of discrete TAO steps. Each step is tightly coupled to the previous one, which limits the model's ability to reorganize its plan or make global optimizations. This structure makes ReAct more prone to local errors and brittle when reasoning requires backtracking or dynamic reorganization.

### A.3  MODEL ERRORS

These examples illustrate how LLMs perform on individual questions and the types of reasoning errors that can occur, even with advanced models.

CASE 1: EDGE CASE — ZONES WITH 0 COMPETITORS ARE EXCLUDED

**User Query:** "I want to open a clothing store in White Plains, with the top category being Other Schools and Instruction and sub-category Exam Preparation and Tutoring. Show me zones with less than 3 competitors in the same category."

**Analysis:** Claude3Haiku filters only among existing competitors, excluding zones with 0 competitors. DeepseekV3 ensures all zones are evaluated by initializing the count to 0.

**Incorrect – Claude3Haiku**

```
category_filtered = filter_pois_by_sub_category(
    poi_spend_df, "Other Schools and Instruction")
competitor_counts = category_filtered['zone_id'].value_counts().
    reset_index()
competitor_counts.columns = ['zone_id', 'num_competitors']
valid_zones = competitor_counts[
    competitor_counts['num_competitors'] < 3]['zone_id'].tolist()
```

**Correct – Deepseek V3**

```
all_zones = poi_spend_df['zone_id'].unique()
category_filtered = filter_pois_by_sub_category(
    poi_spend_df, "Other Schools and Instruction")
competitor_counts = category_filtered['zone_id'].value_counts().to_dict()
zone_to_count = {zone_id: competitor_counts.get(zone_id, 0)
                 for zone_id in all_zones}
valid_zones = [zone_id for zone_id, count in zone_to_count.items() if
    count < 3]
```

CASE 2: PROMPT MISINTERPRETATION

**User Query:** "I want to open a new restaurant, but I need a location with at least 50 parking spots nearby."

**Analysis:** Gemini 1.5 misinterprets "50 parking spots" as requiring 50 different lots, while OpenAI 4o correctly interprets the query as needing one lot with sufficient capacity.

**Incorrect – Gemini 1.5**

```
if get_num_parking(parking_df) >= 50:
    valid_zones.append(zone)
```

**Correct – OpenAI 4o**

```
if get_largest_parking_capacity(parking_df) >= 50:
    valid_zones.append(zone)
```

CASE 3: CODE ERROR

**User Query:** "Looking to build a spa — find me areas where sub-category Advertising Agencies dominates at least 40% of 2024 spend or has 2+ parking spots."

**Analysis:** Deepseek R1 crashes due to lack of error handling in low-data zones, causing a divide-by-zero error. GPT-4o correctly calls the tool and includes a guard clause to ensure stability.

**Incorrect – Deepseek R1**

```
ad_pois = filter_pois_by_sub_category(zone_pois, "Advertising Agencies")
ad_spend = get_spendparam_years(ad_pois, "RAW_TOTAL_SPEND", 2024)
total_spend = 0
if ad_spend / total_spend >= 0.4 or parking_count >= 2:
    valid_zones.add(zone_id)
```

**Correct – GPT-4o**

```
ad_pois = filter_pois_by_sub_category(zone_pois, "Advertising Agencies")
ad_spend = get_spendparam_years(ad_pois, "RAW_TOTAL_SPEND", 2024)
total_spend = get_total_spend(zone_id, 2024)

if (total_spend > 0 and ad_spend / total_spend >= 0.4) or parking_count
    >= 2:
    valid_zones.add(zone_id)
```

These examples illustrate how LLMs perform on individual questions and the types of reasoning errors that can occur, even with advanced LLMs. We have added these case studies to the revised paper to provide more concrete examples of model performance and the challenges faced in this domain.

## A.4 PROMPT ABLATION

To investigate whether prompt design can improve accuracy and reasoning ability, we conduct a prompt ablation study by adding explicit guidance to the instruction. We augment the prompt with phrases such as "think holistically," "consider edge cases," and "do not discard zones early." This modification is evaluated on hard queries using the Gemini model family. The revised prompts are referred to as v2. As shown in Table 5, both Gemini 1.5 and Gemini 2.5 exhibit performance improvements with the revised prompting strategy. The perfect pass rate for Gemini 2.5 increases from 45.10% to 49.02%, a relative improvement of 8.7%, while Gemini 1.5 has a relative improvement of 4.9%. Similar trends are observed across precision, recall, and F1 score, with Gemini 2.5 showing especially notable gains. These findings suggest that better prompting can meaningfully enhance agent reasoning in complex queries where logical missteps and premature filtering are common failure modes. The pronounced improvement on Gemini 2.5 illustrates that stronger models are more responsive to strategic instruction tuning.

Table 4: Performance metrics on LocationReasoner on New York and Tampa

| Location | Model | Difficulty | Delivery Rate | Perfect Pass | Precision | Recall | F1 Score |
|---|---|---|---|---|---|---|---|
| New York | Gemini 1.5 | Simple | 55.40% | 46.87% | 0.35 | 0.28 | 0.31 |
| | | Medium | 49.24% | 34.49% | 0.22 | 0.20 | 0.21 |
| | | Hard | 38.97% | 20.13% | 0.10 | 0.07 | 0.08 |
| | Gemini 2.5 | Simple | 96.73% | 78.88% | 0.67 | 0.63 | 0.65 |
| | | Medium | 89.02% | 73.61% | 0.53 | 0.47 | 0.50 |
| | | Hard | 73.62% | 58.40% | 0.36 | 0.32 | 0.34 |
| | ReAct | Simple | 87.16% | 45.54% | 0.48 | 0.38 | 0.42 |
| | | Medium | 81.05% | 33.72% | 0.41 | 0.19 | 0.26 |
| | | Hard | 78.24% | 15.15% | 0.28 | 0.08 | 0.12 |
| | ReAct + Reflexion | Simple | 98.97% | 46.28% | 0.58 | 0.32 | 0.41 |
| | | Medium | 93.41% | 29.03% | 0.55 | 0.27 | 0.36 |
| | | Hard | 92.37% | 23.95% | 0.35 | 0.14 | 0.20 |
| Tampa | Gemini 1.5 | Simple | 50.48% | 39.12% | 0.29 | 0.27 | 0.20 |
| | | Medium | 53.76% | 44.07% | 0.21 | 0.18 | 0.19 |
| | | Hard | 34.66% | 23.82% | 0.12 | 0.06 | 0.08 |
| | Gemini 2.5 | Simple | 90.21% | 71.26% | 0.62 | 0.66 | 0.64 |
| | | Medium | 84.85% | 67.89% | 0.50 | 0.48 | 0.49 |
| | | Hard | 77.03% | 61.74% | 0.31 | 0.37 | 0.34 |
| | ReAct | Simple | 68.45% | 46.78% | 0.42 | 0.39 | 0.40 |
| | | Medium | 76.12% | 33.25% | 0.33 | 0.21 | 0.26 |
| | | Hard | 59.88% | 18.92% | 0.20 | 0.09 | 0.12 |
| | ReAct + Reflexion | Simple | 96.88% | 35.62% | 0.50 | 0.30 | 0.37 |
| | | Medium | 91.23% | 39.05% | 0.46 | 0.26 | 0.33 |
| | | Hard | 89.47% | 26.73% | 0.28 | 0.12 | 0.16 |

Table 5: Performance metrics on hard queries across Gemini model versions.

| Model | Delivery Rate | Perfect Pass | Precision | Recall | F1 Score |
|---|---|---|---|---|---|
| Gemini 2.5 | 83.33% | 45.10% | 0.3300 | 0.3500 | 0.3200 |
| Gemini 2.5 v2 | 93.14% | 49.02% | 0.3792 | 0.3663 | 0.3437 |
| Gemini 1.5 | 46.08% | 19.61% | 0.1300 | 0.0800 | 0.0900 |
| Gemini 1.5 v2 | 43.14% | 20.58% | 0.1402 | 0.0901 | 0.0927 |

