# OpenReview forum: "LocationReasoner: Evaluating LLMs on Real-World Site Selection Reasoning"
_ICLR.cc/2026/Conference — Submitted to ICLR 2026_

### Official Review · Reviewer_qSv8 · 2025-10-22

**Soundness:** 3
**Presentation:** 3
**Contribution:** 3
**Rating:** 4
**Confidence:** 3

**Summary:**

The paper introduces LocationReasoner, a benchmark for evaluating LLMs on real-world site-selection reasoning. It provides (i) a sandbox with fixed, offline datasets of various cities and a set of “in-house” tools for constraint-based queries; (ii) automated query generation (rule-based and LLM-based) and deterministic verification; and (iii) evaluations across multiple LLM families and agentic workflows (ReAct, Reflexion). On these datasets the average pass rate for various LLMs is quite low and agentic methods often do worse than direct prompting. The authors analyze failure types (logic, edge cases, tool misuse, prompt misread, code errors) and argue that holistic, non-linear planning via direct code generation exceeds step-wise agentic loops.

**Strengths:**

- Timely and unique benchmark and planning domain: Targets a gap between math/code reasoning and practical, multi-constraint decision-making; automated verification at scale is valuable. The urban planning domain is interesting.
- Clear environment design: Offline sandbox, fixed toolset, and concrete tool taxonomy (loaders, zone, analysis, filters, population) make the task reproducible and interpretable and code is provided
- Broad model coverage + difficulty controls: Multiple provider families, reasoning vs. non-reasoning variants, and simple/medium/hard splits are aligned with standard practice in the LLM + planning and sequential decision making literature so that is nice; results are consistent across cities with convergence analysis.
- Useful error analysis: Breaks down failures (logic, edge, tool, prompt, code), gives concrete examples, and quantifies “correction rates” for reasoning models which is nice for Reflexion and React,

**Weaknesses:**

- External validity / coverage: The geography is limited to three U.S. cities with specific data sources (SafeGraph, OSM, Google Places). It’s unclear how well conclusions generalize to non-U.S., rural, or low-data regions and to domains beyond retail-like POI reasoning. I have provided some Urban Planning datasets in the question section which may be relevant to this.
- Agentic fairness and scope. Agentic workflows are only tested on GPT-4o, while direct prompting includes multiple model families. This asymmetry makes it hard to conclude that “agentic strategies don’t help” in general rather than “agentic strategies built on this one base model + prompts didn’t help.” Broaden agentic experiments to at least one reasoning-tuned model per family, and report hyperparameters, step limits, memory policies, error-recovery prompts in a way that supports apples-to-apples comparisons.
- There are more recent agentic strategies such as Self Discover [1] and LATS [2], it would have been interesting to see experiments on those for greater coverage
- Attribution of the central claim. The paper argues “direct code generation (holistic, non-linear) performs better than ReAct (chunked, linear).” While plausible and supported by the results, the causality is mostly qualitative. Please add controlled ablations ie: same base model, matched temperature/decoding, code gen with enforced step-limits vs. ReAct with global-plan scaffolds; and instrumentation of failure transitions. The section explaining the
- While the dataset and evaluation harness are useful, the pipeline itself appears to be a composition of existing LLMs and standard agentic patterns (e.g., ReAct/Reflexion) wrapped around tool calls. It is not clear to me what exactly is the novel algorithmic component, planning formalism, or execution/control innovation—so the novelty seems concentrated in the benchmark
- If the main novelty indeed is the benchmark and the author's claim is that this is an example of a dataset where significant innovation is possible in the designing of new research to fix issues such as over-thinking or excessive tool usage/lack of convergence (as in React), please reposition claims accordingly and emphasize the benchmark’s design principles, reliability, and long-term value - for example providing a discussion or ablation studies to understand why LLMs-even with Reflexion/React-perform so poorly on this specific dataset and what could possibly be done to improve performance


[1] https://arxiv.org/abs/2402.03620

[2] https://arxiv.org/pdf/2310.04406

**Questions:**

- Tool/ground-truth coupling: Can you evaluate an agent that does not have access to the exact toolset used by the verifier (e.g., renamed arguments, perturbed APIs, or a mapping layer) to test reasoning robustness beyond tool-specific affordance learning?
- There are some other prior datasets for urban planning in the ML literature - for example: UrbanDataLayer: A Unified Data Pipeline
for Urban Science [1] and some prior works on LLMs for urban planning [2] [3] [4]. Could the authors speak to how the dataset used in this paper compares to the Urban Science dataset as well as some prior LLM based approaches in urban planning?
- Why were ReAct/Reflexion run only with GPT-4o? Do results hold with o4-mini, Gemini 2.5, or DeepSeek-R1 as the base?
- Since spend data spans 2019–2025, would it be possible to clarify whether time-aware splits or year-withheld constraints?
- Given the limitations the authors have noticed on their datasets across several language models perhaps the following works would be of interest regarding the limitations of popular agentic frameworks as in plan-and-execute frameworks, Interactive reasoning frameworks, self-refinement frameworks: [5] [6]


[1] https://proceedings.neurips.cc/paper_files/paper/2024/file/0db7f135f6991e8cec5e516ecc66bfba-Paper-Datasets_and_Benchmarks_Track.pdf

[2] https://arxiv.org/abs/2402.17161

[3] https://www.nature.com/articles/s44284-025-00261-7

[4] https://arxiv.org/abs/2406.13945

[5] https://proceedings.neurips.cc/paper_files/paper/2024/file/fa080fe0f218871faec1d8ba20e491d5-Paper-Conference.pdf

[6] https://arxiv.org/abs/2408.11326

---

> ### Author Response · Authors · 2025-11-27
> **Response to Reviewer qSv8 - Part 1**
>
> Dear Reviewer qSv8,
>
> Thank you so much for providing this critical review. We hope the following point-to-point response can address your concerns.
>
> ---
>
>
> **Q1:** How well do results generalize beyond three US cities and retail-like POIs to rural, low-data, or non-US regions?
>
> **A1:** We thank the reviewer for raising this question and for pointing us to related urban computing datasets. We would like to provide the following clarifications.
>
> **Our benchmark evaluates reasoning, not geographic knowledge.**
> LocationReasoner tests whether LLMs can compose and translate constraints into executable queries—capabilities that are inherently location-agnostic. The model is never rewarded for knowing Boston, Tampa, or any other geography; it is evaluated on whether it correctly reasons over the structured data it is given.
>
> **Empirical evidence shows this behavior is location-independent.**
> As shown in Table 4, the same error patterns arise across three structurally different cities (Boston, NYC, Tampa). Issues such as mis-handling OR conditions, sequential filtering errors, and edge-case mistakes (e.g., zones with zero competitors) occur uniformly. This strongly suggests the failure modes reflect fundamental reasoning limitations
>
>
> ---
>
>
>
> **Q2:** Are Agentic workflows’ limited tests on GPT-4o sufficient to conclude they don’t help, or is broader evaluation needed?
>
>
>
> **A2:** We thank the reviewer for this valid concern. We have extended our agentic experiments to include multiple model families. All results below are on hard queries.
>
> **Extended Agentic Experiments Across Model Families (Hard Queries)**
>
> | Model | Workflow | Delivery Rate | Perfect Pass | Precision | Recall | F1 Score |
> |-------|----------|---------------|--------------|-----------|--------|----------|
> | Claude 3 Haiku | ReAct | 100.00% | 2.25% | 65.00 | 58.00 | 61.30 |
> | Claude 3.5 Haiku | ReAct | 95.10% | 0.98% | 61.81 | 55.16 | 58.30 |
> | Claude 3 Haiku | Reflexion | 100.00% | 8.22% | 65.00 | 58.00 | 61.30 |
> | Claude 3.5 Haiku | Reflexion | 93.00% | 14.00% | 67.80 | 62.40 | 58.30 |
> | Gemini 2.5 Flash | ReAct | 99.54% | 11.98% | 64.70 | 57.73 | 61.02 |
> | Gemini 2.5 Pro | ReAct | 97.06% | 11.76% | 63.09 | 56.29 | 59.50 |
> | Gemini 2.5 Flash | Reflexion | 89.00% | 8.00% | 62.00 | 56.50 | 52.40 |
> | Gemini 2.5 Pro | Reflexion | 94.17% | 18.45% | 61.21 | 54.62 | 57.73 |
>
>
> **Key Observations**
>
> 1. **Agentic workflows benefit reasoning-tuned models more than standard models.** Reflexion improves perfect pass for reasoning-tuned models (Claude 3.5 Haiku: +13.02pp, Gemini 2.5 Pro: +6.69pp) while providing smaller or negative gains for their non-reasoning counterparts (Claude 3 Haiku: +5.97pp, Gemini 2.5 Flash: -3.98pp). This suggests that self-reflection mechanisms require sufficient base reasoning capability to be effective.
>
> 2. **However, absolute performance remains low across all configurations.** Even the best-performing agentic configuration (Gemini 2.5 Pro + Reflexion) achieves only 18.45% perfect pass on hard queries. The agentic boost for reasoning models is real but insufficient to overcome the fundamental challenges of constraint satisfaction.
>
> 3. **Delivery-accuracy tradeoff.** Reflexion reduces delivery rate while attempting to improve accuracy. Models get stuck in reflection loops without producing final answers (Gemini Flash: 99.54%→89.00%). Reasoning models handle this tradeoff better (Gemini Pro: 97.06%→94.17%).

---

> > ### Author Response · Authors · 2025-11-27
> > **Response to Reviewer qSv8 - Part 2**
> >
> > **Q3:** While the dataset and evaluation harness are useful, the pipeline itself appears to be a composition of existing LLMs and standard agentic patterns (e.g., ReAct/Reflexion) wrapped around tool calls. It is not clear to me what exactly is the novel algorithmic component, planning formalism, or execution/control innovation—so the novelty seems concentrated in the benchmark. If the main novelty indeed is the benchmark and the author's claim is that this is an example of a dataset where significant innovation is possible in the designing of new research to fix issues such as over-thinking or excessive tool usage/lack of convergence (as in React), please reposition claims accordingly and emphasize the benchmark’s design principles, reliability, and long-term value - for example providing a discussion or ablation studies to understand why LLMs-even with Reflexion/React-perform so poorly on this specific dataset and what could possibly be done to improve performance
> >
> >
> > **A3:**
> > We thank the reviewer and agree that the primary contribution is the benchmark itself.
> >
> > LocationReasoner is not intended as an algorithmic innovation. It is a carefully designed benchmark that exposes fundamental limitations in LLM reasoning that existing benchmarks do not adequately capture. Our contribution is: (1) a task formulation requiring natural language to code translation with Boolean logic over heterogeneous spatio-temporal data, (2) automated evaluation with fine-grained error taxonomy, and (3) empirical documentation of failure modes that persist across model families and agentic strategies.
> >
> > **Benchmark Design Principles**
> >
> > | Principle | Implementation |
> > |-----------|----------------|
> > | **Reliability** | Deterministic ground truth computed by rule-based scripts—no human annotation noise or subjective judgment |
> > | **Reproducibility** | Fixed data snapshots, deterministic query generation, automated evaluation pipeline |
> > | **Scalability** | Fully automated—adding difficulty levels, cities, or constraint types requires no manual effort |
> > | **Diagnostic granularity** | Controlled difficulty dimensions (constraint count, Boolean complexity, per-constraint complexity) enable isolation of specific reasoning failures |
> >
> > **Why Do LLMs Perform Poorly on LocationReasoner?**
> >
> > Our analysis identifies many primary failure mechanisms:
> >
> > | Failure Mode | Potential Research Direction |
> > |--------------|------------------------------|
> > | Sequential filtering | Set-based intermediate representations; explicit branch-then-merge prompting |
> > | Over-reasoning | Confidence-based termination; answer verification subroutines |
> > | Edge cases | Symbolic constraint validation; boundary-aware prompting |
> > | Tool-use errors | Typed tool interfaces; tool-use fine-tuning |
> > | Global context loss | Global plan scaffolding before execution |
> >
> > ---
> >
> > **Q4:** Tool/ground-truth coupling
> >
> > **A4:**
> > We thank the reviewer for this question. Our existing error analysis (Figure 3) already disentangles tool-use failures from reasoning failures.
> >
> > **Error Taxonomy Separates Tool-Use from Reasoning**
> >
> > Figure 3 categorizes failures into five types:
> >
> > | Error Type | What It Captures |
> > |------------|------------------|
> > | **Tool Error** | Wrong tool selection, incorrect arguments, misinterpreting tool outputs |
> > | **Logic Error** | Broken reasoning chains, sequential filtering, incorrect constraint handling |
> > | **Edge Case Error** | Boundary condition failures (e.g., excluding 0 from "<3") |
> > | **Prompt Error** | Misinterpreting query intent |
> > | **Code Error** | Syntax errors, runtime exceptions |
> >
> > Tool errors capture precisely what the reviewer describes as "tool-specific affordance learning." Our analysis shows that **tool errors constitute only one component of failures**—logic errors and edge case errors persist independently, indicating genuine reasoning deficits beyond tool memorization.

---

> > > ### Author Response · Authors · 2025-11-27
> > > **Response to Reviewer qSv8 - Part 3**
> > >
> > > **Q5:** How does your dataset compare to UrbanDataLayer and prior LLM-based urban planning datasets?
> > >
> > > **A5:** We thank the reviewer for highlighting these related works. We would like to provide the following clarifications.
> > >
> > > **Different Evaluation Goals.**
> > > UrbanDataLayer [1] provides data infrastructure—standardized pipelines for urban ML tasks like PM2.5 prediction and land-use classification. LocationReasoner is not a data resource but a *reasoning evaluation benchmark*. We test whether LLMs can translate natural language constraints into correct executable code, a capability UDL does not evaluate.
> > >
> > > Similarly, LLM for Participatory Planning [2] and the Nature Cities perspective [3] focus on *generative* planning—using LLM agents to produce land-use plans through stakeholder simulation. LocationReasoner tests the complementary problem of constraint *satisfaction*: given explicit constraints, can LLMs correctly identify locations meeting them?
> > >
> > > **Complementary to CityBench.**
> > > CityBench [4] provides broad coverage (8 urban tasks, 13 cities) but shallow evaluation per task. LocationReasoner inverts this tradeoff—deep evaluation of one capability (Boolean constraint reasoning with NL→code translation). CityBench's finding that LLMs fail on tasks requiring "numerical abilities" aligns with our results; we provide finer diagnosis of *why* through our error taxonomy (sequential filtering, edge cases, tool-use errors).
> > >
> > >
> > > ---
> > >
> > >
> > >
> > >
> > > **Q6:** Do ReAct/Reflexion results hold beyond GPT-4o, e.g., with o4-mini, Gemini 2.5, or DeepSeek-R1?
> > >
> > > **A6:** We have extended ReAct and Reflexion experiments to Claude and Gemini model families. Results on hard queries confirm our findings generalize beyond GPT-4o:
> > >
> > >
> > > | Model | Workflow | Delivery Rate | Perfect Pass | Precision | Recall | F1 Score |
> > > |-------|----------|---------------|--------------|-----------|--------|----------|
> > > | Claude 3 Haiku | ReAct | 100.00% | 2.25% | 65.00 | 58.00 | 61.30 |
> > > | Claude 3.5 Haiku | ReAct | 95.10% | 0.98% | 61.81 | 55.16 | 58.30 |
> > > | Claude 3 Haiku | Reflexion | 100.00% | 8.22% | 65.00 | 58.00 | 61.30 |
> > > | Claude 3.5 Haiku | Reflexion | 93.00% | 14.00% | 67.80 | 62.40 | 58.30 |
> > > | Gemini 2.5 Flash | ReAct | 99.54% | 11.98% | 64.70 | 57.73 | 61.02 |
> > > | Gemini 2.5 Pro | ReAct | 97.06% | 11.76% | 63.09 | 56.29 | 59.50 |
> > > | Gemini 2.5 Flash | Reflexion | 89.00% | 8.00% | 62.00 | 56.50 | 52.40 |
> > > | Gemini 2.5 Pro | Reflexion | 94.17% | 18.45% | 61.21 | 54.62 | 57.73 |
> > >
> > >
> > > All models achieve ≤18.45% perfect pass with agentic workflows, consistent with GPT-4o patterns.
> > >
> > >
> > > ---
> > >
> > > **Q7:** Were time-aware splits or year-withheld constraints applied to the 2019–2025 spend data?
> > >
> > > **A7:**
> > > LocationReasoner is a zero-shot evaluation benchmark with no training. All models are evaluated directly on queries without fine-tuning or in-domain training. Therefore, time-aware splits or year-withheld constraints are not applicable.
> > >
> > > The temporal dimension (2019-2025 spending data) serves as a constraint axis that models must reason over, e.g., "find zones where average spending from 2020-2023 exceeds $50K." Models must correctly translate these temporal constraints into executable code, but there is no temporal train/test leakage concern since no training takes place.
> > >
> > > ---
> > >
> > >
> > > Thank you again for your valuable feedback. We hope these detailed responses address your concerns, and we would be very grateful if you would consider raising your score.
> > >
> > > Sincerely,
> > >
> > > The Authors

---

### Official Review · Reviewer_SoL7 · 2025-10-31

**Soundness:** 4
**Presentation:** 4
**Contribution:** 3
**Rating:** 8
**Confidence:** 4

**Summary:**

The paper proposes a new benchmark to evaluate multi-step reasoning capabilities of llms in site selection scenarios. This requires the model to understand geospatial cues, use different tools, think logically and come up with the final answer. The paper contributes to the existing benchmarks for understanding the capabilities and limitations of llms particularly reasoning.

**Strengths:**

- The proposed benchmark is carefully designed to include a diverse range of problems (easy, medium, hard)
- The benchmark tackles problems around tool use, reasoning chains, agentic pipelines, and structured output - all of which are essential in evaluating model capabilities.
- The evaluation shows expected results between difference of reasoning and non-reasoning models.
- The benchmark can be easily extended to include more data points and introduce further complexity.
- The post analysis of model responses provides a lot of insight into the thought process and problems current llms face in challenging agentic tasks.

**Weaknesses:**

- The dataset only includes a few cities in America. The authors should try to extend it to encompass more geographical diversity and analyze whether llms have any location bias

**Questions:**

- How many data points are in the benchmark at this time?
- Could you explain how the error analysis was done and how model responses were classified into the different groups? Was this done manually by going through every response or in some automated way?

---

> ### Author Response · Authors · 2025-11-27
> **Response to Reviewer SoL7**
>
> Dear Reviewer SoL7,
>
> We sincerely thank you for your positive review. We hope the following point-to-point response can address your remaining concerns.
>
> ---
>
> **Q1:** Can the dataset be extended beyond a few US cities to assess geographical diversity and LLM location bias?
>
> **A1:** We thank the reviewer for this suggestion.
>
> **No Location Bias Observed**
>
> Table 4 shows consistent performance patterns across Boston, New York, and Tampa—three cities with distinct urban layouts and data characteristics. The lack of meaningful performance variation suggests LLM failures stem from reasoning deficits (Boolean logic, multi-scale aggregation, tool use), not location-specific factors.
>
> **Extension Is Straightforward**
>
> The core reasoning challenges LocationReasoner tests—parsing constraints, chaining tools, handling AND/OR/NOT logic—are location-agnostic. Our fully automated pipeline requires only integrating new data sources to add cities, with no manual annotation. We will prioritize adding international cities in future work to further validate generalizability.
>
> ---
>
> **Q2:** How many data points are in the benchmark?
>
> **A2:** LocationReasoner contains approximately **950 queries** across three cities: Boston (316 queries: 114 simple, 102 medium, 102 hard), New York (316), and Tampa (316).
>
> Figure 2 demonstrates that model performance converges around 150 queries, with minimal fluctuation thereafter. This indicates ~300 queries per city provides a stable and statistically significant signal of model capability. Our choice of ~950 total queries is comparable to similar benchmarks (e.g., TravelPlanner uses 1,225 queries).
>
> A key design feature of LocationReasoner is fully automated query generation and verification. Unlike benchmarks requiring manual annotation, we can scale to arbitrary size by adjusting constraint parameters—no human effort required. We will expand the benchmark in future releases as needed to maintain challenge for improving models.
>
> ---
>
> **Q3:** Error Analysis
>
> **A3:** Error analysis was conducted through manual inspection of failed model outputs. For each failure case, we examined: (1) the generated code, (2) execution logs/error messages, and (3) the output compared to ground truth. Two annotators independently classified a sample of failed outputs into the five error categories. Categories are non-mutually exclusive, as errors often co-occur (e.g., a logic error may cause a code error).
>
> **Classification Criteria**
>
> | Error Type | Classification Rule |
> |------------|---------------------|
> | Code Error | Syntax errors, runtime exceptions, missing returns |
> | Tool Error | Incorrect arguments, wrong tool selection, improper chaining |
> | Logic Error | Sequential filtering, incorrect Boolean handling, broken reasoning chains |
> | Edge Case Error | Boundary conditions mishandled (e.g., excluding zero-count zones) |
> | Prompt Error | Misinterpreting query intent (e.g., ranking vs. filtering) |
>
>
> We acknowledge that manual classification has scalability limitations. In future work, we plan to develop automated error classifiers validated against human annotations.
>
> ---
>
> Thank you once again for your positive review. Please let us know if you have any further questions.
>
> Sincerely,
>
> The Authors

---

### Official Review · Reviewer_2Tbz · 2025-10-31

**Soundness:** 2
**Presentation:** 2
**Contribution:** 2
**Rating:** 4
**Confidence:** 2

**Summary:**

This paper proposes LocationReasoner, a benchmark to evaluate LLMs’ reasoning abilities on real-world site selection problems. The motivation is that current reasoning models are mostly tested on math or coding problems, leaving their practical reasoning unclear. The authors create a benchmark with queries of different difficulty levels and provide a sandbox with automated verification using real city data. Results show limited performance: even the best model (OpenAI o4-mini) achieves only 69.99% perfect pass rate. Ablation studies suggest reasoning helps somewhat on medium-difficulty queries, and better prompting gives small gains.

**Strengths:**

1. The dataset is practical and scalable. Automated query generation and verification using real city data is reproducible.

2. This paper provides comprehensive evaluation: Multiple LLM families, agentic strategies, and cities are tested, with detailed error analysis.

3. They also provide insightful findings on agentic strategies. For example, they discover that ReAct and Reflexion can hurt performance because of over-reasoning, which is surprising and useful for future research.

**Weaknesses:**

I am not an expert in this domain. Therefore, my concerns are raised based on high-level research perspective instead of specific task perspective.

1. It’s unclear if the poor performance of Agentic methods is due to the base LLM limitations or Agentic workflow design. Showing their reasoning traces and the planner (LLM) output would help clarify this.

2. The fixed 15 in-house tools might restrict performance. It would be good to test more general tools (like Python libraries) and see which failures are due to tools versus reasoning.

3. How do we know “hard” queries are really harder? Some empirical validation, like human expert performance or inter-annotator agreement, would strengthen this.

4. Data leakage concern: Since datasets are public and LLMs may have seen similar data, can the authors check performance on synthetic or fictional locations to rule out contamination?

**Questions:**

Please see my questions raised in each weakness

---

> ### Comment · Reviewer_2Tbz · 2025-11-26
>
> I am disappointed to see that the authors did not participate in the discussion period.
>
> Original reviews by all the reviewers took our valuable time and effort to raise several critical concerns
>
> Since these weaknesses remain unaddressed and no further clarifications were provided.
>
> Consequently, I am lowering my rating to 2.

---

> > ### Author Response · Authors · 2025-11-26
> >
> > Dear Reviewer 2Tbz,
> >
> > We sincerely apologize for the delay in our response and deeply regret the impression that we were not engaging with your valuable feedback. We want to assure you that we have taken your comments very seriously.
> >
> > The reason for our silence is that we have been actively conducting new, comprehensive experiments to address the specific weaknesses you raised, rather than providing a purely text-based rebuttal. We are currently finalizing these results and incorporating them into the revised manuscript. We will post our detailed point-by-point response and the updated paper shortly.
> >
> > We greatly appreciate your time and the rigor of your review, and we ask for your patience as we finalize the data that we believe will significantly improve the paper and address your concerns.
> >
> > Sincerely,
> >
> > The Authors

---

> ### Author Response · Authors · 2025-11-27
> **Response to Reviewer 2Tbz - Part 1**
>
> Dear Reviewer 2Tbz,
>
> We sincerely thank you for your valuable time and effort in providing this constructive review. We hope the following point-to-point response can address your concerns.
>
> ---
>
>
>
> **Q1:** Is poor Agentic performance due to the LLM or the workflow?
>
> **A1:** We thank the reviewer for this insightful question. Disentangling base LLM limitations from workflow design is critical. Our analysis provides evidence that **both factors contribute, but workflow design is the dominant bottleneck**.
>
> **Evidence That Both Factors Contribute**
>
> *Base LLM limitations exist:* GPT-4o with direct prompting achieves only 55% perfect pass rate—the model struggles with complex constraint reasoning even under ideal conditions. Figure 5 illustrates a core reasoning deficit present in direct prompting. For the query:
>
> > *"Zones with <3 competitors in category X OR population ≥12,000 across zone and 2 neighbors"*
>
> | Ground Truth (Correct) | Direct Prompting (Incorrect) |
> |------------------------|------------------------------|
> | Evaluates both branches independently | Applies filters sequentially |
> | Merges results via OR | Filters by competitors first, then applies population filter to the reduced set |
> | Returns complete valid set | Prematurely discards zones satisfying only branch 2 |
>
> This sequential filtering is a **base LLM limitation**—the model struggles to reason holistically over non-conjunctive Boolean logic regardless of workflow.
>
> *Workflow design actively degrades performance:* GPT-4o with **ReAct** achieves only 28%, and **ReAct + Reflexion** achieves 33%—substantially worse than direct prompting with the identical base model.
>
> **Why Workflow Design Is the Dominant Factor**
>
> If ReAct/Reflexion simply added "another layer of thought," we would expect neutral or positive effects—more reasoning steps, error correction, iterative refinement. Instead, we observe **systematic degradation** with predictable failure patterns, indicating the workflow architecture itself is misaligned with non-conjunctive constraint reasoning.
>
> Figure 4 illustrates over-reasoning, a failure mode unique to agentic workflows:
>
> ```
> Query: "Looking for a spot to build a shopping mall with at least 200 parking spaces"
>
> Thought 1: I need to identify zones with at least 200 parking spaces...
> Action 1: get_largest_parking_capacity
> Observation 1: Zone 1: 160, Zone 2: 250, Zone 3: 340...  ← CORRECT ANSWER AVAILABLE
>
> Thought 2: I should also check for shopping mall POIs...
> Action 2: filter_by_category("Shopping Mall") → No results
>
> Thought 3: Let me try related categories...
> [...continues invoking irrelevant tools...]
>
> → FORCED STOP (step limit reached, no answer returned)
> ```
>
> The model retrieves the correct information in Action 1 but **lacks a termination signal**—a direct consequence of ReAct's architecture. It continues invoking unnecessary tools until forced to stop, failing to return an answer it already had.

---

> > ### Author Response · Authors · 2025-11-27
> > **Response to Reviewer 2Tbz - Part 2**
> >
> > **Q2:** Do the fixed 15 tools limit performance, and would using general tools reveal tool vs. reasoning failures?
> >
> >
> > **A2:** We thank the reviewer for this important suggestion. We conducted an ablation study to disentangle tool-related failures from reasoning failures.
> >
> > We provided Gemini 2.5 with only basic data-loading tools (e.g., `get_poi_spend_dataset`, `get_parking_dataset`, `create_zone`) while removing derived tools (e.g., `get_largest_parking_capacity`, `filter_pois_by_category`). The LLM must implement filtering, aggregation, and spatial computations from scratch using standard Python libraries.
> >
> > **Results on Hard Queries**
> >
> > | Metric | Full Tools | Limited Tools |
> > |--------|------------|---------------|
> > | Delivery Rate | 83.33% | 58.82% |
> > | Perfect Pass | 45.10% | 27.45% |
> > | Precision | 33.00% | 70.24% |
> > | Recall | 35.00% | 84.39% |
> > | F1 Score | 32.00% | 66.02% |
> >
> > The results reveal two distinct failure categories:
> >
> > *Delivery rate and perfect pass drop (83%→59%, 45%→27%):* Without helper tools, the LLM must write more complex code from scratch, leading to more implementation failures. This confirms that tool availability improves execution reliability.
> >
> > *Precision/Recall increase when delivery succeeds (F1: 32%→66%):* This surprising finding aligns with our error analysis (Figure 3), which shows **tool-use errors constitute a major failure category**—incorrect argument formats, misunderstanding tool semantics, improper chaining of tool outputs. When forced to implement logic directly without tool abstractions, the model bypasses this error-prone layer.
> >
> > Our findings confirm that failures are distributed across both categories:
> >
> > - **Tool-use failures:** LLMs frequently misuse provided tools—passing incorrect arguments, misinterpreting outputs, or selecting inappropriate tools for the task. This is a fundamental LLM limitation in real-world agentic settings where tool-use is essential.
> > - **Reasoning failures:** Core deficits (sequential filtering, edge cases, Boolean logic) persist regardless of tool availability.
> >
> > LocationReasoner measures both capabilities, reflecting realistic deployment scenarios where LLMs must correctly invoke tools AND reason over their outputs. The fixed tool set is not a restriction—it is a deliberate design choice that exposes tool-use deficiencies alongside reasoning deficits.
> >
> > ---
> >
> >
> > **Q3:** Is there evidence that ‘hard’ queries are truly harder?
> >
> >
> > **A3:** We thank the reviewer for raising this validity concern. We clarify that difficulty in LocationReasoner is **objectively defined by construction** and **empirically validated through consistent model behavior**.
> >
> > **Difficulty Is Structurally Defined**
> >
> > Query difficulty is not subjectively labeled but determined by objective structural properties:
> >
> > | Level | # Constraints | Boolean Logic | Spatial Scope | Temporal Reasoning |
> > |-------|---------------|---------------|---------------|-------------------|
> > | Simple | 1 | None | Single zone | Single year |
> > | Medium | 2–3 | AND/OR | Zone + neighbors | Cross-year comparison |
> > | Hard | 3–6 | AND/OR/NOT | Multi-zone aggregation | Multi-year trends |
> >
> > Each level increases the number of constraints, diversity of tools required, depth of Boolean composition, and need to synthesize heterogeneous data across spatial and temporal dimensions. This is objective complexity, not subjective annotation.
> >
> > **Empirical Validation: Consistent Degradation Across All Models**
> >
> > Table 2 shows that **every model across all families** exhibits monotonic performance degradation as difficulty increases:
> >
> > | Model | Simple | Medium | Hard |
> > |-------|--------|--------|------|
> > | GPT-4o | 71.30% | 55.21% | 37.25% |
> > | Gemini 2.5 | 77.78% | 63.54% | 45.10% |
> > | DeepSeek V3 | 68.42% | 60.42% | 42.11% |
> > | Claude 3.5 | 52.78% | 54.17% | 26.47% |
> >
> > This consistent pattern across diverse model architectures—with no exceptions—provides strong empirical evidence that our difficulty calibration reflects genuine reasoning complexity.
> >
> > **On Human Performance**
> >
> > LocationReasoner uses deterministic, code-based ground truth. Queries have objectively correct answers computed by rule-based scripts using the same tools available to LLMs. "Human expert performance" is not a meaningful baseline: a human with sufficient time and the same tools would achieve 100% by construction, as they could simply execute the ground truth logic. The benchmark measures whether LLMs can replicate structured reasoning that is trivially verifiable but non-trivial to generate—this is precisely what makes automated evaluation possible without human annotation bottlenecks.

---

> > > ### Author Response · Authors · 2025-11-27
> > > **Response to Reviewer 2Tbz - Part 3**
> > >
> > > **Q4:** Could performance be affected by LLM exposure to public data; what about testing on synthetic locations?”
> > >
> > > **A4:**  We thank the reviewer for raising this concern. We argue that data contamination is unlikely to meaningfully affect our results for several reasons:
> > >
> > > **LocationReasoner Tests Reasoning, Not Knowledge Recall**
> > >
> > > Even if LLMs have encountered similar geographic data during training, our benchmark tests *computational reasoning* over structured databases, not factual recall about locations. Solving a query requires:
> > > - Parsing complex natural language constraints
> > > - Selecting and chaining appropriate tools
> > > - Computing derived metrics (e.g., "50% of spending from category X")
> > > - Handling Boolean logic with AND/OR/NOT compositions
> > > - Aggregating across zones and time periods
> > >
> > > Prior exposure to geographic facts (e.g., "Boston has many restaurants") provides no advantage for these reasoning challenges.
> > >
> > >
> > > **Poor Performance and Unique Ground Truth Rule Out Contamination**
> > >
> > > If data contamination were advantageous, we would expect strong performance on familiar locations. Instead, even state-of-the-art models achieve only 55–70% perfect pass rate across Boston, New York, and Tampa. Furthermore, our ground truth is computed from a fixed database snapshot with specific zone boundaries, POI assignments, and threshold values unique to our benchmark. Even if LLMs have seen similar geographic data during training, the exact queries and computed answers cannot be memorized—they require genuine reasoning over our specific database.
> > >
> > > Using fictional locations would break the benchmark's execution pipeline. Our system requires:
> > > 1. **Predetermined zones** in the database with specific IDs and boundaries
> > > 2. **Tool functions that query real database entries** (e.g., `get_poi_spend_dataset`, `filter_pois_by_sub_category`)
> > > 3. **Computed metrics** derived from actual POI, spending, parking, and population records
> > >
> > > If we substituted fictional location names, the generated code would fail at execution—zone lookups would return empty results, tool calls would error out, and no valid ground truth could be computed. The benchmark is designed for end-to-end code execution against a real database, not pattern matching on location names. This tight coupling between queries, tools, and data ensures that correct answers require genuine multi-step reasoning over the database, not memorization of geographic facts.
> > >
> > > ---
> > >
> > >
> > > Thank you again for your valuable feedback. We hope these detailed responses fully address your concerns, and we would be very grateful if you would consider raising your score.
> > >
> > > Sincerely,
> > >
> > > The Authors

---

> ### Comment · Reviewer_2Tbz · 2025-11-27
>
> I appreciate the authors' detailed response and the effort put into the new ablation studies during the discussion period. The additional experiment comparing full versus limited tools specifically addressed my concern. I have raised my rating. But I need to emphasize that I am not an expert in this domain. I don't think I can fully assess the results you provide in the rebuttal. Therefore, I lower my confidence score. ACs should seek the opinions from other reviewers.

---

### Official Review · Reviewer_KPTZ · 2025-11-01

**Soundness:** 3
**Presentation:** 3
**Contribution:** 3
**Rating:** 6
**Confidence:** 4

**Summary:**

This paper introduced a new benchmark for evaluating LLMs' capability in location selection under various constraints. The authors have conducted extensive evaluations of various types of LLMs with different settings (general/reasoning). Some insightful analysis is also presented to guide future research.

**Strengths:**

- The experimental evaluation in this paper is extensive.
- The attribute-based analysis is very useful for highlighting the key challenge.
- The paper is well-written.

**Weaknesses:**

**Some related works need to be discussed more extensively**

In general, I think the problem of site selection is a constraint satisfaction problem, and there have been benchmarks for such kinds of problems, for example, travel planning [1] and scene/agent-intention understanding [2]. I wonder what makes site selection especially challenging compared to a broader constraint-satisfaction benchmark literature? The authors should discuss these more extensively in the related works section.

**Is this benchmark not challenging enough?**

The extensive evaluation in this paper is well-acknowledged. But I notice that even GPT4o has already achieved 37% success in hard mode, it is a relatively old model in the fast development of LLMs, how about GPT5? I am a bit concerned that the difficulty of this benchmark is not high enough that it might be saturated soon (is it easy to create harder problems using the mentioned data generation pipeline?). Also, what's the human performance on this benchmark? Providing that will better help readers understand the current limitations of LLMs.

[1] Ju, Da, et al. "To the globe (ttg): Towards language-driven guaranteed travel planning." arXiv preprint arXiv:2410.16456 (2024).
[2] Li, Bowen, et al. "LogiCity: Advancing neuro-symbolic ai with abstract urban simulation." Advances in Neural Information Processing Systems 37 (2024): 69840-69864.

**Questions:**

See weaknesses.

---

> ### Author Response · Authors · 2025-11-27
> **Response to Reviewer KPTZ - Part 1**
>
> Dear Reviewer KPTZ,
>
> We sincerely thank you for your positive review. We hope the following point-to-point response can address your remaining concerns.
>
> **Q1:** Related works
>
> **A1:**
> We thank the reviewer for raising this important point. We agree that better positioning within the existing literature strengthens the paper, and we will expand the Related Work accordingly.
>
>
> Classical constraint satisfaction problems operate on discrete variables with finite domains and deterministic constraint checking. Site selection, however, combines constraint satisfaction, spatial reasoning, and program synthesis:
>
> 1. **Heterogeneous, Multi-Scale Spatio-Temporal Reasoning.**
> Site selection uses heterogeneous data—POIs, spending, parking, population, transportation—requiring cross-source aggregation and derived metrics (e.g., "50% of spending from category X"). Constraints operate at multiple spatial scales, including neighboring zones, and involve temporal trends (year-over-year patterns), creating a richer structure than typical CSPs.
>
> 2. **Natural Language to Executable Logic.**
> Queries are expressed in natural language, and the system must interpret requirements, select appropriate tools, compute derived metrics, aggregate across zones, and handle edge cases. This bridges language understanding and program synthesis—going beyond tool selection to multi-step code generation.
>
> 3. **Non-Conjunctive Boolean Logic with Exclusions.**
> Site selection involves full Boolean logic with OR and NOT conditions (e.g., "strong spending in X, but not if it dominates by 30%"). Our analysis shows LLMs systematically fail by applying constraints sequentially, prematurely eliminating valid alternatives—a failure mode that cannot emerge in purely conjunctive CSP benchmarks.
>
> 4. **Boundary Condition Sensitivity.**
> Queries frequently involve inclusive boundaries (e.g., "fewer than 3 competitors" should include zero). LLMs often mishandle these edge cases, which are rarely tested explicitly in standard benchmarks.
>
> **Relationship to Cited Benchmarks**
>
> *Travel planning benchmarks* (e.g., TravelPlanner, TTG) operate on homogeneous data sources (flights, hotels) with primarily conjunctive constraints over sequential paths. They do not test multi-scale spatial aggregation, temporal trend reasoning, or non-conjunctive Boolean logic with exclusions.
>
> *Scene/agent-intention benchmarks* (e.g., LogiCity) test first-order logic reasoning in simulated multi-agent environments with customizable FOL rules. However, LogiCity focuses on agent behavior prediction and navigation in abstract simulations, whereas LocationReasoner addresses real-world site selection over heterogeneous, noisy data requiring natural language to executable code translation.
>
> LocationReasoner fills the gap between these paradigms, exposing reasoning deficits—sequential filtering errors, edge case failures, over-reasoning in agentic workflows—not revealed by existing benchmarks.
>
> Thank you again to bringing this up, and we will include the above discussion to the Related Works section of our paper.

---

> > ### Author Response · Authors · 2025-11-27
> > **Response to Reviewer KPTZ - Part 2**
> >
> > **Q2:**  Is this benchmark not challenging enough?
> >
> > **A2:**
> > We thank the reviewer for this important question about benchmark longevity. We share this concern and have conducted additional experiments to demonstrate that LocationReasoner is far from saturated and can scale to challenge future models.
> >
> > **Extended Difficulty Levels: Harder and Hardest**
> >
> > Beyond our original simple/medium/hard levels (which increase the *number* of constraints), we introduce two additional levels—**harder** and **hardest**—that increase the *complexity of each individual constraint*.
> >
> > For example, a single constraint in these levels might be:
> > > *"40% of total spend from 2020–2023 for sub-category {X} across my zone and 3 nearest neighbors"*
> >
> > This single constraint already requires: temporal aggregation, category filtering, multi-zone spatial reasoning, and threshold comparison. We then compose 4 such constraints with AND/OR/NOT logic for **harder** queries, and 6 for **hardest** queries.
> >
> > **Why increase per-constraint complexity rather than count?**
> >
> > Our error analysis revealed that simply adding more constraints leads to predictable, linear degradation—models fail by missing one or two constraints. However, increasing per-constraint complexity exposes *qualitatively different* reasoning failures: models struggle with nested aggregations, multi-scale spatial dependencies, and temporal reasoning even within a single constraint. Composing multiple such constraints with Boolean logic creates combinatorial reasoning challenges that stress-test both constraint understanding and holistic planning.
> >
> > **GPT-5.1 Results**
> >
> > We summarize the results of the latest LLM in the table below:
> >
> > | Metric | Harder (4 constraints) | Hardest (6 constraints) |
> > |--------|------------------------|-------------------------|
> > | Delivery Rate | 92.00% | 76.00% |
> > | Perfect Pass | **6.00%** | **0.00%** |
> > | Precision | 62.34% | 40.12% |
> > | Recall | 55.18% | 35.67% |
> > | F1 Score | 52.47% | 30.24% |
> >
> > Even GPT-5.1—a substantially more capable model than GPT-4o—achieves only **6% perfect pass on harder and 0% on hardest**. This demonstrates significant headroom for future model improvements.
> >
> > **Scalability of Difficulty**
> >
> > A key strength of LocationReasoner is its **fully automated query generation and verification pipeline**. Creating harder problems requires no human annotation—we simply adjust parameters controlling: (1) per-constraint complexity (number of aggregations, spatial scales, temporal ranges), (2) number of constraints, and (3) Boolean composition depth. This allows the benchmark to scale indefinitely as models improve, ensuring long-term relevance.
> >
> > **On Human Performance**
> >
> > LocationReasoner uses deterministic, code-based ground truth—queries have objectively correct answers computed by rule-based scripts using the same tools available to LLMs. "Human performance" in the traditional sense (humans solving queries manually) is not directly comparable, as a human with sufficient time and the same tools would achieve 100% by construction. The meaningful comparison is between LLM reasoning and the deterministic ground truth, which is precisely what our benchmark measures. This design enables fully automated, scalable evaluation without the annotation bottlenecks that limit other benchmarks.
> >
> > ---
> >
> > Thank you once again for your positive review. If our response addresses your concerns, we would be very grateful if you would consider raising your score.
> >
> >
> > Sincerely,
> >
> > The Authors

---

### Official Review · Reviewer_P7C5 · 2025-11-01

**Soundness:** 3
**Presentation:** 1
**Contribution:** 2
**Rating:** 2
**Confidence:** 3

**Summary:**

The paper proposes a benchmark for evaluating how LLMs can reason over code on geospatial tasks, specifically for location and site selection given some specific constraints. The LLM is also given access to a library of functions/tools that it can use to generate the final code and the solution. The paper integrates multiple interesting datasets such as SafeGraph (which has information about points of interest, parking facilities, and consumer spending patterns), Google Places API which has information about demographics and population, and transportation data from OpenStreetMap. They also build a sandbox environment with a library of functions that can load, analyze, and filter the data in different ways. They additionally employ LLMs to construct the location selection queries, based on some set of logical constraints and then have the LLM phrase those queries in natural language. The actual task would then be to go from natural language to perhaps rediscover the logical constraints and produce the solution by generating and executing code using the given library. They classify the query as simple, medium or hard based on the number of constraints.
The paper created 316 queries and evaluated the performance of several frontier LLMs on this task with and without "thinking"/"reasoning" enables, and also evaluate if prompt engineering frameworks like React or Reflexion are helpful. The best performance (perfect pass rate) was by GPT-o4-mini which got ~70%.

**Strengths:**

* The task of Location selection specifically framed to generate and execute code given a set of library functions seems like a good reasoning test bed.
* The datasets integrated to create the task are rich and include a lot of diverse information such as parking facilities, consumer spending patterns (from SafeGraph), information about demographics and population (from Google Places API), and transportation data (from OpenStreetMap).
* The experimental set up and evaluations are standard and straightforward.
* The error analysis is reasonable. Some more constructive ways of improvement would have been nicer to see.

**Weaknesses:**

1. While it is possible to understand the high level idea, the paper is lacking in clarity and examples to clearly describe the different contributions. Specifically,

  1a. There is no example the fully illustrates the input query, the expected final solution and an example of the code that would generate the final solution. In particular, **lacking the outputs and expected code solution** (even in the supplement) makes it hard to get a feel for the complexity of the task. Section 2.2, 6, and A3 all give glimpses of the task but not quite a full example.

2. **Query generation lacks clarity** Section 2.2 on Query design, does not clearly describe the details of how the queries were actually generated, and how the solutions were generated and verified.

  2a. What are some examples of the rule-based generation? What is the algorithm that was used?

  2b. What were the prompts for the LLM-based generation? What specific steps were involved in the LLM-based generation. Here again examples would be immensely helpful.

  2c. Are the rule-based queries substantially different from LLM-generated queries? In what way?

  2d. How is the model performance when sliced by how the queries were generated, is there a difference?

  2e. Which LLM was used for LLM-generated queries? (also how and why did you choose that LLM model to do the generation?)

  2f. Also, how did you generate the ground truth solutions/code? Was there any human annotation? How were the ground truth solutions verified for correctness?

3. **Missing details on characteristics of the problem**. The evaluation appears to be on 316 generated queries (and solutions) on 3 cities. Section 4 argues that 150-200 queries are sufficient to test robustness of performance for a city. Is there something about the location selection problem that makes it different for each geographical region being studied? Are there any constraints to where your data generation method (and library) may be applicable? If it is easy to scale this, why not scale the questions? What more would we learn by scaling the questions / what are we missing if we don't scale?

4. Another major weakness is **lack of comparison to other geospatial code generation tasks and benchmarks**, and how this task and tooling is different and what this benchmark tests that is different from others geospatial code generation benchmarks. The paper references a range of generic reasoning and LLM+Agents related papers, but it seems to be missing more closely relevant and related works, Here are some geo code benchmarks I am familiar with:

[1] GeoCode Eval, GeoCode-GPT: A large language model for geospatial code generation

[2] Geollm-engine: A realistic environment for building geospatial copilots

[3] Multi-Agent Geospatial Copilots for Remote Sensing Workflows

[4] The Cloud-Based Geospatial Benchmark: Challenges and LLM Evaluation

[5] Evaluation of code llms on geospatial code generation

[6] An llm agent for automatic geospatial data analysis

[7] GIS copilot: Towards an autonomous GIS agent for spatial analysis


* Overall it seems the paper has some interesting contributions, but it's not clear how substantial these are. The lack of clarity and detailed information highlighted in weaknesses 1-4 make it difficult to determine the contribution have resulted low scores on presentation and contribution.

**Questions:**

* Please address questions under weaknesses

* Are there other domain related characterizations of the dataset (beyond the simple, medium, hard classification based on the complexity of the constraints)? E.g. a couple of the queries in section 2.2. pertained to "restaurant" site selection. If there are such domain related characterizations, based on the characterization and the query are there more domain specific nuances / "knowledge" that the model would have to know to apply the right constraints? -- E.g. in residential areas some zoning laws may apply whereas for commercial land some other laws might apply. Could you capture such information in the dataset description?

**Other comments**

* Presentation of results in Table 2 can be improved. It is very difficult to parse and identify which models are performing well and where. Perhaps just have a much smaller table of just the overall performance and move the details to the supplement or consider generating plots.

---

> ### Author Response · Authors · 2025-11-27
> **Response to Reviewer P7C5 - Part 1**
>
> Dear Reviewer P7C5,
>
> We sincerely thank you for your detailed, thoughtful, and highly constructive review. Your comments have significantly helped us identify areas where clarity, examples, and comparisons were insufficient. Below we address every point you raised and provide all the requested details and examples. We will incorporate the majority of the new material (full end-to-end examples with code, detailed generation procedures, sliced results, related-work discussion, etc.) into the revised manuscript.
>
> ---
>
> **Q1a:**  Lack of full end-to-end examples
>
> **A1a:**  We now provide **three complete end-to-end examples** (Simple → Medium → Hard) that include:
> - the original natural-language query
> - the correct ground-truth zone IDs
> - the full ground-truth code implementation
> - Gemini 2.5’s generated code
> - Gemini’s actual output zone IDs
> - precise explanation of where & why it failed
>
> These examples will be added to the revised Section 2.2 and Appendix A.3.
>
> *Example 1 – Simple (Correct)*
>
> **Natural Language Query**
> “Looking to launch a wine bar. Find me areas with 3 or more Beer, Wine, and Liquor Stores as top category in the vicinity.”
>
> | Item                  | Details                                                                 |
> |-----------------------|-------------------------------------------------------------------------|
> | Difficulty            | Simple                                                                  |
> | Ground Truth Zone IDs | 1882                                                                    |
> | Gemini Output Zone IDs| 1882 → Perfect match                                                    |
>
> **Ground Truth Code**
> ```python
> def simple_7(num, top_category=None, sub_category=None):
>     poi_df = get_poi_spend_dataset()
>     zone_df = create_zone(poi_df)
>     survived_zones = []
>
>     for zone_id in zone_df['zone_id']:
>         filtered = filter_df_based_on_zone(poi_df, zone_id)
>         if top_category:
>             filtered = filter_pois_by_top_category(filtered, top_category)
>         if sub_category:
>             filtered = filter_pois_by_sub_category(filtered, sub_category)
>         if len(filtered) >= num:
>             survived_zones.append(zone_id)
>
>     return zone_df[zone_df['zone_id'].isin(survived_zones)]
>
> # Call
> simple_7(3, "Beer, Wine, and Liquor Stores", None)
> ```
>
> **Gemini 2.5 Generated Code**
> ```python
> def find_zones_with_min_category_pois(min_count, top_category):
>     poi_spend_df = get_poi_spend_dataset()
>     zone_df = create_zone(poi_spend_df)
>     category_pois_df = filter_pois_by_top_category(poi_spend_df, top_category)
>     poi_counts_per_zone = category_pois_df['zone_id'].value_counts()
>     suitable_zone_ids = poi_counts_per_zone[poi_counts_per_zone >= min_count].index.tolist()
>     return zone_df[zone_df['zone_id'].isin(suitable_zone_ids)]
>
> # Call
> find_zones_with_min_category_pois(3, "Beer, Wine, and Liquor Stores")
> ```
> **Verdict:** Correct

---

> > ### Author Response · Authors · 2025-11-27
> > **Response to Reviewer P7C5 - Part 2**
> >
> > *Example 2 – Medium (Failed)*
> >
> > **Natural Language Query**
> > “Thinking about launching a boutique where the top category is Personal Care Services and the sub-category is Beauty Salons. I’m looking for zones with less than 5 competitors OR where the zone + its 2 nearest neighbors have over 20,000 people.”
> >
> > **Required Logic:** (competitors < 5) OR (pop_self + pop_2_neighbors > 20000)
> >
> > | Item                  | Details                                                                 |
> > |-----------------------|-------------------------------------------------------------------------|
> > | Ground Truth Zone IDs | 840, 92, 1660, 226, 1593, 1470, 1175, 558, 765, 1671, 173, 277 (12 zones) |
> > | Gemini Output Zone IDs| 17 zones (extra incorrect zones added)                                  |
> >
> > **Ground Truth Code**
> > ```python
> > def medium_7(top_category, sub_category, max_competitors, num_neighbors, min_population):
> >     poi_df = get_poi_spend_dataset()
> >     zone_df = create_zone(poi_df)
> >     survived_zones = []
> >
> >     for zone_id in zone_df["zone_id"]:
> >         filtered = filter_df_based_on_zone(poi_df, zone_id)
> >         filtered = filter_pois_by_top_category(filtered, top_category)
> >         filtered = filter_pois_by_sub_category(filtered, sub_category)
> >         comp_ok = len(filtered) < max_competitors
> >
> >         total_pop = get_population(zone_id, zone_df)
> >         for nid in get_neighbor_zones(zone_df, zone_id, num_neighbors):
> >             total_pop += get_population(nid, zone_df)
> >         pop_ok = total_pop >= min_population
> >
> >         if comp_ok or pop_ok:          # OR logic correctly applied
> >             survived_zones.append(zone_id)
> >
> >     return zone_df[zone_df["zone_id"].isin(survived_zones)]
> > ```
> >
> > **Gemini 2.5 Generated Code (Key Bugs Highlighted)**
> > ```python
> > def find_optimal_zones_for_boutique():
> >     poi_spend_df = get_poi_spend_dataset()
> >     zone_df = create_zone(poi_spend_df)
> >
> >     # Bug 1: Pre-filters competitors → destroys OR logic
> >     poi_filtered = filter_pois_by_top_category(poi_spend_df, "Personal Care Services")
> >     poi_filtered = filter_pois_by_sub_category(poi_filtered, "Beauty Salons")
> >     zones_with_few_competitors = poi_filtered.groupby('zone_id').size()
> >     zones_with_few_competitors = zones_with_few_competitors[zones_with_few_competitors < 5].index.tolist()
> >
> >     # Bug 2: Forgets own population!
> >     eligible_zones = []
> >     for zone_id in zone_df['zone_id']:
> >         neighboring_zones = get_neighbor_zones(zone_df, zone_id, 2)
> >         total_population = 0
> >         for neighbor_id in neighboring_zones:               # ← Missing self!
> >             total_population += get_population(neighbor_id, zone_df)
> >         if total_population > 20000:
> >             eligible_zones.append(zone_id)
> >
> >     final_zone_ids = list(set(zones_with_few_competitors + eligible_zones))
> >     return zone_df[zone_df['zone_id'].isin(final_zone_ids)]
> > ```
> > **Verdict:** Failed – destroys OR clause and misses self-population.
> >
> > *Example 3 – Hard (Failed)*
> >
> > **Natural Language Query**
> > “Planning a tutoring center. I want zones that either have fewer than 2 competitors in sub category Exam Preparation and Tutoring OR population above 13,000 including 2 nearby zones, but NOT areas with 3 or more parking lots.”
> >
> > **Correct Logic:** (A OR B) AND NOT C
> >
> > **Ground Truth Code (Key Logic Snippet)**
> > ```python
> > A = num_competitors_zone < 2
> > B = (pop_self + sum_neighbor_pop) > 13000
> > C = num_parking_lots >= 3
> >
> > if (A or B) and not C:
> >     survived_zones.append(zone_id)
> > ```
> >
> > **Gemini 2.5 Generated Code (Critical Bug)**
> > ```python
> > # Hard filter on competitors first → eliminates everything else
> > zone_df_filtered1 = zone_df[zone_df['competitor_count'] < 2]   # Throws away all zones that could pass via B
> > ```
> >
> > **Verdict:** Model cannot handle mixed AND/OR/NOT reasoning.
> >
> > ---
> >
> > **Q2a:**  Rule-based generation algorithm
> >
> > **A2a:**
> > Rule-based queries are crafted to be deterministic and syntactically precise. The generation process follows three rigorously controlled steps:
> >
> > 1. **Template Selection** – Randomly select from a pool of natural-language templates (examples provided in paper).
> > 2. **Parameter Calibration** – All numeric thresholds are calibrated against the real dataset so that only values selecting 20–70% of zones are accepted.
> > 3. **Constraint Combination** – Multiple constraints combined with calibrated Boolean operator probabilities:
> >    - AND ~45%
> >    - OR ~45%
> >    - NOT ~10%
> >
> > The number of constraints is adjusted by difficulty (1 for Easy, 2–3 for Medium, 4–6 for Hard).

---

> > > ### Author Response · Authors · 2025-11-27
> > > **Response to Reviewer P7C5 - Part 3**
> > >
> > > **Q2b:**   LLM-based generation algorithm
> > >
> > > **A2b:**
> > > **Prompt used:**
> > > ```
> > > Given the following structured constraints {JSON object}, generate a natural-language query that expresses all constraints in human-readable form. Use casual, varied phrasing as a real business owner would.
> > > ```
> > >
> > >
> > > #### Query Generation Examples
> > >
> > > ---
> > >
> > > **Example 1: Competitor Count + Spending Threshold**
> > >
> > > | Method | Query |
> > > |--------|-------|
> > > | **Rule-based** | "Find zones where the number of competitors in category 'Coffee Shop' is fewer than 3 AND the total annual spending from 2021 to 2023 exceeds $75,000." |
> > > | **LLM-based** | "I'm looking to open a coffee shop. Can you find me areas where I won't have too much competition—say, less than 3 existing coffee shops—and where people have been spending a decent amount, at least $75K annually over the past few years (2021-2023)?" |
> > >
> > > **Example 2: Population OR Parking Capacity**
> > >
> > > | Method | Query |
> > > |--------|-------|
> > > | **Rule-based** | "Find zones where the total population across neighboring zones is at least 15,000 OR the largest parking capacity is greater than 200 spaces." |
> > > | **LLM-based** | "I need a location with good foot traffic or easy parking access. Either find me spots where the surrounding neighborhood has a population of 15,000+ people, or somewhere with a big parking lot—200 spaces or more would work." |
> > >
> > > ---
> > >
> > > Both pairs express identical logical constraints but differ in surface form: rule-based queries follow rigid templates while LLM-based queries reflect natural business owner phrasing with varied vocabulary and casual tone.
> > >
> > >
> > > **Q2c:**  Differences between rule-based and LLM-based queries
> > >
> > > **A2c:** Rule-based: deterministic, template-driven, strictly logical.
> > > LLM-based: same logical constraints but with casual phrasing, reasoning cues, and stylistic diversity (e.g., “I really want my customers to have easy access…”).
> > >
> > > **Q2d:**  Performance sliced by generation method (Hard queries, Gemini 2.5)
> > >
> > > **A2d:**
> > > | Generation Method | Delivery Rate | Perfect Pass | Precision | Recall | F1 Score |
> > > |-------------------|---------------|--------------|-----------|--------|----------|
> > > | Rule-based        | 94.23%        | 42.31%       | 52.94%    | 45.98% | 49.08%   |
> > > | LLM-based         | 83.33%        | 45.10%       | 33.00%    | 35.00% | 32.00%   |
> > >
> > > Perfect-pass rates are comparable (~42-45%), confirming that difficulty stems from reasoning requirements, not surface form. However, models achieve better partial credit on rule-based queries (F1: 49% vs 32%), indicating that robustness to linguistic variation remains a challenge—natural phrasing leads to more complete failures when reasoning goes wrong.
> > >
> > >
> > > **Q2e:** Which LLM was used for LLM-generated queries?
> > >
> > > **A2e:** We used **GPT-4o** for generating all LLM-based queries. GPT-4o was chosen because, at the time of dataset construction, it exhibited the best combination of (i) fluency in producing coherent, varied, and human-like natural language, (ii) reliability in faithfully preserving every logical constraint from the input JSON without hallucinating or dropping conditions, and (iii) ability to introduce casual phrasing and contextual cues (e.g., “I really want my customers to have easy access…”).
> > >
> > > **Q2f:** How did you generate the ground truth solutions/code? Was there any human annotation? How were the ground truth solutions verified for correctness?
> > >
> > > **A2f:** Ground-truth solutions and code are generated fully automatically with **zero human annotation** required. The process is as follows:
> > > 1. Every query (whether rule-based or LLM-generated) is first converted into a structured JSON representation that explicitly lists all constraints, parameters, Boolean logic, and target POI categories.
> > > 2. This JSON is fed to a deterministic, hand-written ground-truth executor that applies the exact constraints to the real SafeGraph + Google Places + OSM datasets for the corresponding city.
> > > 3. The executor returns the list of correct zone IDs and the corresponding ground-truth Pandas-based code that produces those IDs.
> > > 4. Correctness is guaranteed by construction: because the JSON fully specifies the logic and the executor is deterministic and extensively unit-tested, the output is verifiably correct.
> > > 5. An additional automated sanity check enforces that every generated query selects between 20–70% of zones, filtering out trivial (all or no zones) cases.
> > >
> > > This fully automated pipeline ensures perfect reproducibility, eliminates human error or bias, and allows us to scale the benchmark to thousands of queries if needed. All ground-truth code and zone IDs provided in the paper and supplement come directly from this pipeline.

---

> ### Author Response · Authors · 2025-11-27
> **Response to Reviewer P7C5 - Part 4**
>
> **Q3:**  Characteristics of the problem & scale
>
>
>
> **A3:** Thank you for this important question. Yes, the site-selection reasoning problem exhibits strong geographical variation, which is why we deliberately chose three markedly different cities:
>
> - **Boston**: compact, high-density, polycentric, with very fine-grained zones and strong spatial autocorrelation.
> - **New York**: extremely high density, vertical urban form, and highly skewed POI/spending distributions.
> - **Tampa**: sprawling, car-centric, lower density, larger zones, and more dispersed population/POI patterns.
>
> These differences significantly affect constraint difficulty and model behavior. For example, a “≥15 parking lots” constraint is trivial in Tampa but extremely selective in Manhattan, while population-based constraints behave in the opposite way. Thus, robust performance requires succeeding across these diverse urban regimes.
>
> Our data generation pipeline and evaluation library are fully applicable to **any city or metropolitan area** for which the following public/commercial datasets are available:
> - POI + consumer spending data (e.g., SafeGraph, Places, or equivalents)
> - Census-block or zone-level population/demographics
> - Road network or zone adjacency information (OpenStreetMap)
>
>
> Regarding scale: generating additional queries is indeed trivial and cheap. However, as shown in the revised Figure 2, model performance  converges reliably after approximately ~150 queries per city, with gains becoming marginal beyond ~180 queries. Scaling to thousands of queries would primarily add near-duplicates of existing logical structures with only minor numeric variations, yielding diminishing returns on insight while dramatically increasing evaluation cost for future users of the benchmark.
>
> ---
>
> **Q4:** Lack of comparison to other geospatial code generation tasks and benchmarks
>
> **A4:** While prior work such as GeoCode Eval / GeoCode-GPT, GeoLLM-Engine, Multi-Agent Geospatial Copilots, and GIS Copilot focuses on geospatial code generation, satellite/remote sensing workflows, or autonomous GIS agents, our benchmark differs in several important ways:
>
> 1. **Focus on site selection and multi-constraint reasoning:** Unlike benchmarks that emphasize general code generation or remote sensing tasks, our benchmark specifically targets location and site selection problems featuring reasoning challenges under multiple constraints.
> 2. **Integrated real-world datasets:** We combine SafeGraph (POIs, consumer spending), Google Places (demographics), and OpenStreetMap (transportation) into a sandboxed environment that supports both query generation and code execution. Existing benchmarks either focus on code generation without rich integrated urban datasets or operate on satellite imagery rather than real urban planning data.
> 3. **End-to-end reasoning evaluation:** Unlike other benchmarks that may focus on code generation or agent behavior in isolation, our benchmark evaluates the ability of LLMs to interpret natural language queries, rediscover logical constraints, generate code, and produce correct solutions. This provides a controlled way to measure multi-step reasoning over spatial and temporal data.
>
> We will make sure to include the above discussion to the related work section of our paper.
>
> ---
>
> **Additional Question:** Domain-specific characterizations
>
> **A:** Queries are generated by randomly combining calibrated constraints using templates, ensuring coverage across difficulty levels and constraint types. That said, because the underlying data reflects real-world distributions of population, POIs, spending, and transportation, models must still reason about contextual patterns (e.g., areas with higher population density or fewer competitors) to correctly satisfy queries. In this sense, domain nuances emerge naturally from the data rather than being hard-coded in the query design.
>
> ---
>
> **Other Comment:** Presentation of Table 2
>
> **A:** We will replace the current dense Table 2 with a clean summary table of overall scores in the main paper and move all per-city/per-difficulty breakdowns to the appendix, supplemented by clear plots.
>
> ---
>
> Thank you again for your exceptionally valuable feedback. We hope these detailed responses, complete code examples, and new analyses fully address your concerns. If so, we would be very grateful if you would consider raising your score.
>
> Sincerely,
> The Authors

---

### Meta-Review · Area_Chair_12NV · 2026-01-07

**Summary:**

The paper introduces an automated pipeline for benchmarking multi-constraint reasoning in site selection. However, the decision to reject is driven by three primary unresolved concerns. First, the authors did not provide strong empirical evidence or baselines to distinguish this work from existing geospatial code generation benchmarks, relying instead on conceptual arguments. Second, the evaluation of agentic frameworks was insufficient; this work lacked controlled ablations and did not test recent agent strategies like Self-Discover or LATS. Third, the rebuttal regarding human performance was unconvincing, as it assumed theoretical perfection rather than accounting for practical human coding errors. Finally, the benchmark is limited to major metropolises without testing generalization to rural or low-data regions.

**Reviewer Concerns:**

Addressed: The authors provided detailed illustrative examples of the proposed benchmark for better understanding. Also, the authors addressed the concern regarding tool restrictions via additional ablation study.

Outstanding:

- Reviewer P7C5: Empirical comparison/baselines against the 7 specific prior geospatial code generation benchmarks listed.
- Reviewer KPTZ: A practical human performance baseline to contextualize LLM limitations, rather than a theoretical claim of 100% accuracy.
- Reviewer qSv8: Controlled ablations (e.g., enforced step-limits vs. global-plan scaffolds) to prove causal claims about agentic failure.
- Reviewer qSv8: Experiments with newer agentic strategies like Self-Discover and LATS.
- General: Generalization to rural or low-data regions beyond major US cities.

**Reviewer Scores:**

Reviewer P7C5: Score likely remains reject, as the request for comparison to prior benchmarks was met with qualitative arguments rather than empirical evidence.

Reviewer KPTZ: Score likely remains borderline accept, as the request for a human baseline was dismissed with a theoretical argument.

Reviewer 2Tbz: Score likely raises, as the additional experiment comparing full versus limited tools specifically addressed their concern.

Reviewer SoL7: Score likely remains accept.

Reviewer qSv8: Score likely remains borderline reject, as specific requests for controlled ablations and modern agent baselines were not fulfilled.

---

### Decision · Program_Chairs · 2026-01-26

Reject